# Towards Multi-dimensional Explanation Alignment for Medical Classification

**Lijie Hu**[*,1,2]**, Songning Lai**[*,1,2,3]**, Wenshuo Chen**[*,1,2]**, Hongru Xiao**[4]
**Hongbin Lin**[3]**, Lu Yu**[6]**, Jingfeng Zhang**[5,7]**, and Di Wang**[1,2]
[1]Provable Responsible AI and Data Analytics (PRADA) Lab
[2]King Abdullah University of Science and Technology
[3]HKUST(GZ)    [4]Tongji University    [5]The University of Auckland
[6]Ant Group, [7]RIKEN Center for Advanced Intelligence Project (AIP)

## Abstract

The lack of interpretability in the field of medical image analysis has significant ethical and legal implications. Existing interpretable methods in this domain encounter several challenges, including dependency on specific models, difficulties in understanding and visualization, as well as issues related to efficiency. To address these limitations, we propose a novel framework called **Med-MICN** (**Med**ical **M**ulti-dimensional **I**nterpretable **C**oncept **N**etwork). Med-MICN provides interpretability alignment for various angles, including neural symbolic reasoning, concept semantics, and saliency maps, which are superior to current interpretable methods. Its advantages include high prediction accuracy, interpretability across multiple dimensions, and automation through an end-to-end concept labeling process that reduces the need for extensive human training effort when working with new datasets. To demonstrate the effectiveness and interpretability of Med-MICN, we apply it to four benchmark datasets and compare it with baselines. The results clearly demonstrate the superior performance and interpretability of our Med-MICN.

## 1 Introduction

The field of medical image analysis has witnessed remarkable advancements, especially for the deep learning models. Deep learning models have exhibited exceptional performance in various tasks, such as image recognition and disease diagnosis [31, 46, 1], with an opaque decision process and intricate network. However, this lack of transparency is particularly problematic in the medical domain, making it challenging for physicians and clinical professionals to trust the predictions made by these deep models. Thus, there is an urgent need for the interpretability of model decisions in the medical domain [43, 13, 57].

The medical field has strict trust requirements. It not only demands high-performing models but also emphasizes comprehensibility and earning the trust of practitioners [20]. Thus, Explainable Artificial Intelligence (XAI) has emerged as a prominent research area in this field. It aims to enhance the transparency and comprehensibility of decision-making processes in deep learning models and large language models by incorporating interpretability [65, 21, 22, 64, 63, 25, 8]. Various methods have been proposed to achieve interpretability, including attention mechanisms [56, 61, 27, 26, 24], saliency maps [70, 16], DeepLIFT and Shapley values [38, 4], influence functions [34, 55]. These methods strive to provide users with visual explanations that shed light on the decision-making process of the model. However, while these post-hoc explanatory methods offer valuable information, there is still a gap between their explanations and the model decisions [42]. Moreover, these post-hoc

---

*The first three authors contributed equally to this work.

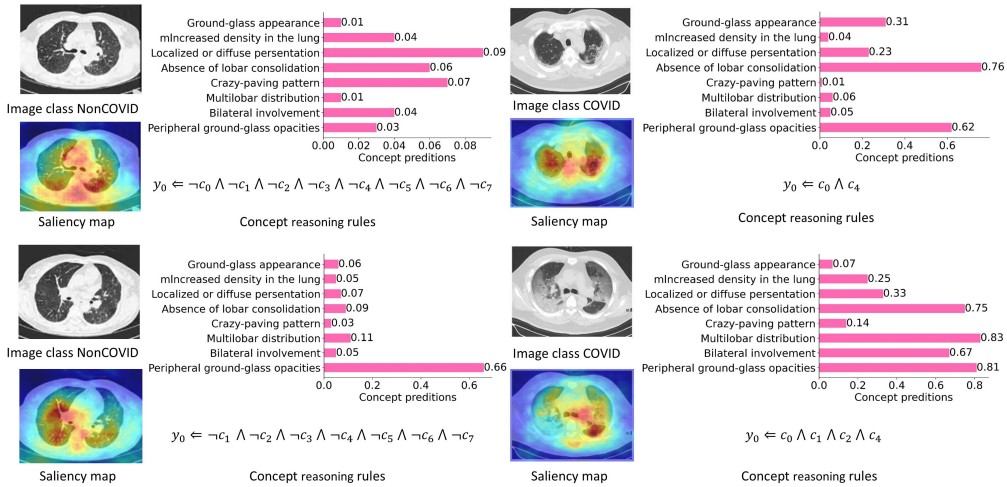

Figure 1: Med-MICN demonstrates multidimensional interpretability, encompassing concept score prediction, concept reasoning rules, and saliency maps, achieving alignment within the interpretative framework. The 'Peripheral ground-glass opacities' is $c_0$, and along the y-axis, it sequentially becomes $c_1, \ldots, c_7$.

explanations are generated after the model training and cannot actively contribute to the model fine-tuning process, hindering them from being a faithful explanation tool.

Thus, there is increasing interest among researchers in developing self-explanatory methods. Among these, concept-based methods have garnered significant attention [50, 2, 35]. Concept Bottleneck Model (CBM) [35] initially predicts a set of pre-determined intermediate concepts and subsequently utilizes these concepts to make predictions for the final output, which are easily understandable to humans. Concept-based explanations provided by inherently interpretable methods are generally more comprehensible than post-hoc approaches. However, most existing methods treat concept features alone as the determination of the predictions. This approach overlooks the intrinsic feature embeddings present within medical images, thus degrading accuracy [44]. Moreover, while these concepts are human-understandable, they lack semantic meanings, thus questioning the faithfulness of their interpretability [39]. To improve further human trust, several recent works [3] aim to leverage syntactic rule structures to concept embeddings. However, there are still several potential issues. First, unlike CBMs, current concept models with logical rules mainly focus on the supervised concept case, which is quite strict for biomedical images as concept annotation is expensive. Second, while current concept models (with logical rules) provide interpretations via concepts, we found that the importance of these concepts is misaligned with other explanations, especially the explanation given by saliency maps [70, 16]. This will lead to a possible reduction in human trust when using these models.

To address these challenges, we introduce a new and innovative end-to-end concept-based framework called the **Med-MICN** (**Med**ical **M**ulti-dimensional **I**nterpretable **C**oncept **N**etwork), as illustrated in Figure 3. As shown in Figure 2, Med-MICN is an end-to-end framework that leverages Large Multimodals (LMMs) to generate concept sets and perform auto-annotation for medical images, thereby aligning concept labels with images to overcome the high cost associated with medical concepts annotation. In contrast to typical concept-based models, our interpretation is notably more diverse and precise (shown in Figure 1). Specifically, we map the image features extracted by the backbone through a concept encoder to obtain concept prediction and concept embeddings, which are then input into the neural symbolic layers for interpretation. This also establishes alignment between image information and concept embeddings by utilizing a concept encoder, leading to the derivation of predictive concept scores. Furthermore, we align concept semantics with concept embeddings by incorporating neural symbolic layers. Thus, we effectively align image information with concept semantics and concept saliency maps, achieving comprehensive multidimensional alignment. Additionally, unlike most concept-based methods, we use concept embeddings to complement the original image features, which enhances classification accuracy without any post-process. Our main contributions can be summarized as follows:

- We proposed an end-to-end framework called Med-MICN, which leverages the strength of different XAI methods such as concept-based models, neural symbolic methods, saliency maps, and concept semantics. Moreover, Med-MICN generates rules and utilizes concept embeddings to complement the intrinsic medical image features, which improves accuracy.

- Med-MICN offers an alignment strategy that includes text and image information, saliency maps, and concept semantics. It is model-agnostic and can easily transfer to other models. Our outputs are interpreted in multiple dimensions, including concept prediction, saliency maps, and concept reasoning rules, making it easier for experts to identify and correct errors.

- Through extensive experiments on four benchmark datasets, Med-MICN demonstrates superior performance and interpretability compared with other concept-based models and the black-box model baselines.

## 2 Related Work

**Concept Bottleneck Model.** The Concept Bottleneck Model (CBM) [35] has emerged as an innovative deep-learning approach for image classification and visual reasoning by incorporating a concept bottleneck layer into deep neural networks. However, CBM faces two significant challenges. Firstly, its performance often falls short of the original models without the concept bottleneck layer, attributed to incomplete information extraction from the original data to bottleneck features. Secondly, CBM extensively depends on meticulous dataset annotation. To solve these problems, researchers have delved into potential solutions. For example, [7] have extended CBM into interactive prediction settings by introducing an interaction policy to determine which concepts to label, ultimately improving the final predictions. Additionally, [41] has addressed the limitations of CBM by proposing a novel framework called Label-free CBM, which offers promising alternatives. Post-hoc Concept Bottleneck models [66] can be applied to various neural networks without compromising model performance, preserving interpretability advantages. Despite much research in the image field [18, 33, 32, 45, 47, 36, 28, 23, 37], concept-based method for the medical field remains less explored, which requires more precise results and faithful interpretation. [9] used a conceptual alignment deep autoencoder to analyze tongue images representing different body constituent types based on traditional Chinese medicine principles. [35] introduced CBM for osteoarthritis grading and used ten clinical concepts such as joint space narrowing, bone spurs, calcification, etc.

However, previous research heavily relies on expert annotation datasets or often focuses solely on concept features to make predictions while overlooking the intrinsic feature embeddings within images. Furthermore, while the concept neural-symbolic model has been explored in the graph domain [3], its application to images, particularly in the medical domain, has been largely absent. Additionally, our work addresses these gaps by proposing an end-to-end framework with an alignment strategy that leverages various explainable methods, including concept-based models, neural-symbolic methods, saliency maps, and concept semantics, to provide comprehensive solutions to these challenges.

**Explanation in Medical Image.** Research on the interpretability of deep learning in medical image processing provides an effective and interactive approach to enhancing medical knowledge and assisting in disease diagnosis. User studies involving physicians have revealed that doctors often seek explanations to understand AI results, especially when the outcomes are related to their own hypotheses or differential diagnoses [60]. They also turn to explanations to resolve conflicts when their judgments differ from those of AI [6], thereby enhancing the intelligence of medical models. Previous studies have visualized lesion areas through methods such as heatmaps [59] and attention visualization [12], aiding in the identification of lesion regions and providing visual evidence. Additionally, utilizing language model-based methods like LLM or LMM to generate medical reports complements the interpretation of model results (ChatCAD [58], XrayGPT [52], Med-PaLM [49]). Saliency maps have emerged as the most common and clinically user-friendly explanation for medical imaging tasks [53, 68, 69]. Recent research underscores the importance of understanding the pivotal features influencing AI predictions, particularly when clinicians must compare AI decisions with their own clinical assessments in cases of decision incongruity [54]. In addition to the image's intrinsic feature recognition, assisted discrimination methods based on concept injection are widely employed in assisted medical diagnosis [35, 7]. Compared to relying solely on self-supervised training, conceptual feature-based supplementation integrates expert knowledge, offering more accurate assistance for interpreting detection results.

However, previous research on medical images relies on single-dimensional explanations, potentially lacking sufficient decision information for physicians. Furthermore, erroneous single-dimensional explanations could significantly impact physicians' judgments. Thus, there is a pressing need for a multi-dimensional explanatory framework where explanations across various dimensions complement each other. In instances of incorrect explanations, physicians can turn to explanations in alternative dimensions to aid their judgment.

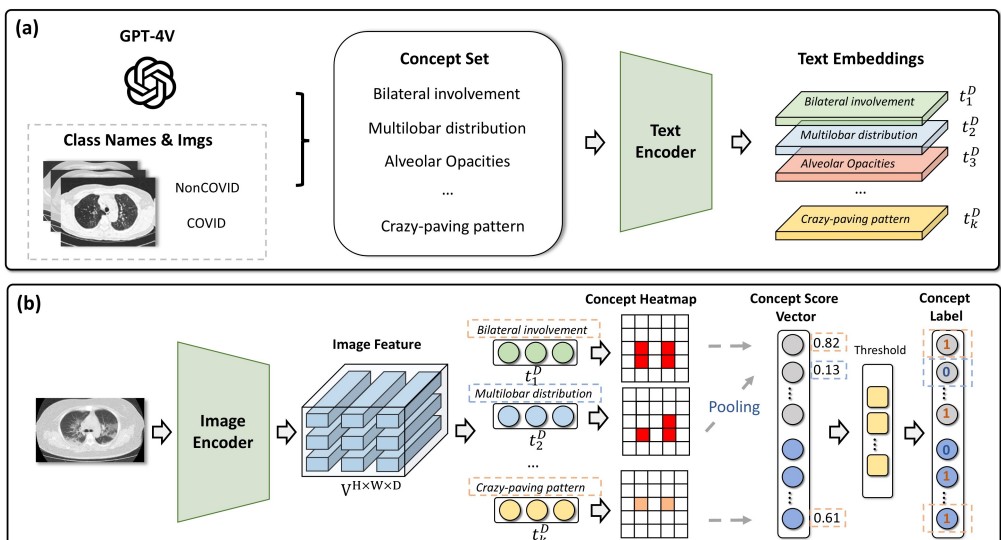

Figure 2: (a) module, output rich dimensional interpretable conceptual information for the specified disease through the multimodal model and convert the conceptual information into text vectors through the text embedding module; (b) module, access the image to the image embedder to get the image features, and then match with the conceptual textual information to get the relevant attention region. Then, we get the influence score of the relevant region information through pooling, and finally send it to the filter to sieve out the concept information with weak relevance to get the disease concept of image information.

## 3 Preliminaries

**Concept Bottleneck Models.** To introduce the original Concept Bottleneck Models, we adopt the notations used by [35]. We consider a classification task with a concept set denoted as $\mathcal{C} = C_1, C_2, \ldots, C_N$ and a training dataset represented as $\{(x_i, y_i, c_i)\}_{i=1}^M$. Here, for $i \in [M]$, $x_i \in \mathbb{R}^d$ represents the feature vector, $y_i \in \mathbb{R}^{d_z}$ denotes the label (with $d_z$ corresponding to the number of classes), and $c_i \in \mathbb{R}^{d_c}$ represents the concept vector. In this context, the $j$-th entry of $c_i$ represents the weight of the concept $p_j$. In CBMs, our goal is to learn two representations: one that transforms the input space to the concept space, denoted as $g : \mathbb{R}^d \to \mathbb{R}^{d_c}$, and another that maps the concept space to the prediction space, denoted as $f : \mathbb{R}^{d_c} \to \mathbb{R}^{d_z}$. For any input $x$, we aim to ensure that its predicted concept vector $\hat{c} = g(x)$ and prediction $\hat{y} = f(g(x))$ are close to their underlying counterparts, thus capturing the essence of the original CBMs.

**Fuzzy Logic Rules.** As described by [17, 3], continuous fuzzy logic extends upon traditional Boolean logic by introducing a more nuanced approach to truth values. Rather than being confined to the discrete values of either 0 or 1, truth values are represented as degrees within the continuous range of $\{0, 1\}$. Conventional Boolean connectives including t-norm $\wedge : [0, 1] \times [0, 1] \mapsto [0, 1]$, t-conorm $\vee : [0, 1] \times [0, 1] \mapsto [0, 1]$, negation $\neg x = 1 - x$. The logical connectives, including $\neg, \vee, \wedge, \Rightarrow, \Leftarrow, \Leftrightarrow$, are utilized to convey the logical relationships between concepts and their representations. For example, consider the problem of deciding whether an X-ray lung image has COVID, given the vocabulary of concepts "ground-glass opacities (GO)," "Localized or diffuse presentation (LDP)," and "lobar consolidation (LC)." A simple decision rule can be $y \Leftrightarrow c_{GO} \wedge \neg c_{LC}$. From this rule, we can deduce that (1) Having both "no LC" and "GO" is relevant to having COVID. (2) Having LDP is irrelevant to deciding whether COVID exists.

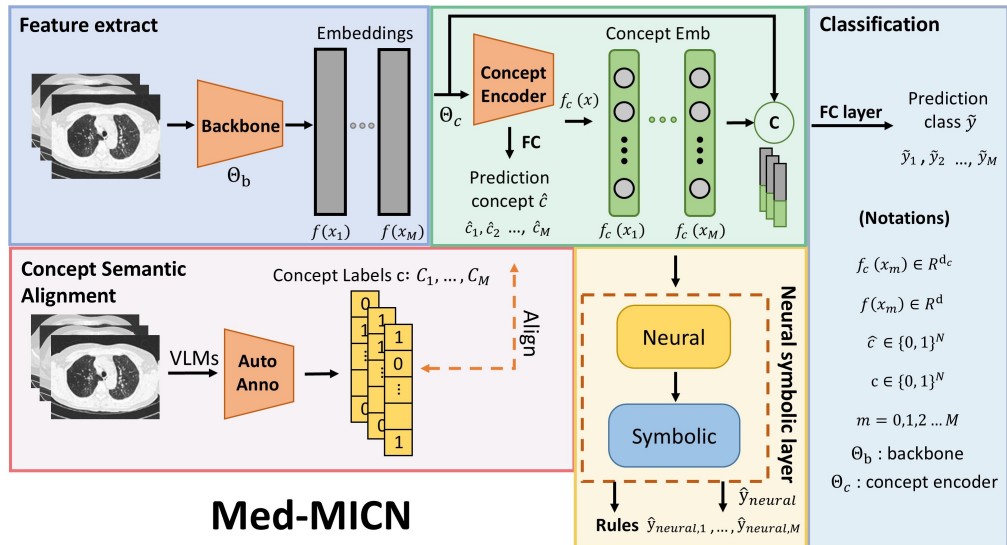

Figure 3: Overview of the Med-MICN Framework. The Med-MICN framework consists of four primary modules: (1) **Feature Extraction Module**: In the initial step, image features are extracted using a backbone network to obtain pixel-level features. (2) **Concept Embedding Module**: The extracted features are fed into the concept embedding module. This module outputs concept embeddings while passing through a category classification linkage layer to obtain predicted category information. (3) **Concept Semantic Alignment**: Concurrently, a Vision-Language Model (VLM) is used to annotate the image features, generating concept category annotations aligned with the predicted categories. (4) **Neural Symbolic Layer**: After obtaining the concept embeddings, they are input into the Neural Symbolic layer to derive conceptual rules. Finally, the concept embeddings obtained from module (2) are concatenated with the original image embeddings and fed into the final category prediction layer to produce the ultimate prediction results.

## 4 Medical Multi-dimensional Interpretable Concept Network

Here, we present Med-MICN (Figure 3), a novel framework that constructs a model in an automated, interpretable, and efficient manner. (i) In traditional CBMs, the concept set is typically generated through annotations by human experts. When there is no concept set and concept labels, we first introduce the automated concept labeling alignment process (Figure 2). (ii) Then, the concept set (output by LLMs such as GPT4-V) is fed into the text encoder to obtain word embedding vectors. Our method utilizes Vision-Language Models (VLMs) to encode the images and calculate cosine distances to generate heatmaps. We apply average pooling to these heatmaps to obtain a similarity score aligned with the concept set through a threshold to obtain concept labels. (iii) Next, we extract image features using a feature extraction network and then map them through a concept encoder to obtain concept embeddings. (iv) We finally use these concept embeddings as input into the neural symbolic layers to generate concept reasoning rules and incorporate them as complementary features to the intrinsic spatial features for predictions, proving multi-dimensional interpretation alignment. We provide details for each component of Med-MICN as follows.

### 4.1 Concept Set Generation and Filtering

Given $M_c$ classes of target diseases or pathology, the first step of our paradigm is to acquire a set of useful concepts related to the classes. A typical workflow in the medical domain is to seek help from experts. Inspired by the recent work, which suggests that instruction-following large language models present a new alternative to automatically generate concepts throughout the entire process [41, 40]. We propose to generate a concept set using LMMs, such as GPT-4V, which has extensive domain knowledge in both visual and language, to identify the crucial concepts for medical classification. Figure 2 (a) illustrates the framework for concept set generation. Details are in Appendix A.

### 4.2 VLMs-Med-based Concept Alignment

**Generating Concept Heatmaps.** Suppose we have a set of $N$ useful concepts $\mathcal{C} = \{C_1, C_2, \ldots, C_N\}$ obtained from GPT-4V. Then, the next step is to generate pseudo-(concept)

labels for each image in the dataset corresponding to its concept set. Inspired by the satisfactory performance in concept recognition within the medical field demonstrated by VLMs [62], we use BioViL [5] to generate the pseudo-labels for the concepts of each image. Figure 2 (b) illustrates the detailed automatic annotation process of concepts.

Given an image $x$ and a concept set, its feature map $V \in \mathbb{R}^{H \times W \times D}$ and the text embedding $t_i \in \mathbb{R}^D$ for each concept are extracted as follows:

$$V = \Theta_V(x), \qquad t_i = \Theta_T(c_i), \quad i = 1, \ldots, N$$

where $\Theta_V$ and $\Theta_T$ are the visual and text encoders, $t_i$ is the embedding of the $i$-th concept in the concept pool, $H$ and $W$ are the height and width of the feature map.

Given $V$ and $t_i$, we can obtain a heatmap $P_i$, i.e., a similarity matrix that measures the similarity between the concept and the image can be obtained by computing their cosine distance:

$$P_{h,w,i} = \frac{t_i^T V_{h,w}}{||t_i|| \cdot ||V_{h,w}||}, \quad h = 1, \ldots, H, \quad w = 1, \ldots, W$$

where $h, w$ are the $h$-th and $w$-th positions in the heatmaps, and $P_{h,w,i}$ represents a local similarity score between $V$ and $t_i$. Then, we derived heatmaps for all concepts, denoted as $\{P_1, P_2, \ldots, P_N\}$.

**Calculating Concept Scores.** As average pooling performs better in downstream medical classification tasks [62], we apply average pooling to the heatmaps to deduce the connection between the image and concepts: $s_i = \frac{1}{H \cdot W} \sum_{h=1}^H \sum_{w=1}^W H_{h,w,i}$. Intuitively, $s_i$ is the refined similarity score between the image and concept $c_i$. Thus, a concept vector $e$ can be obtained, representing the similarity between an image input $x$ and a set of concepts: $e = (s_1, \ldots, s_N)^T$.

**Alignment of Image and Concept Labels.** To align images with concept labels, we determine the presence of a concept attribute in an image based on a threshold value derived from an experiment. If the value $s_i$ exceeds this threshold, we consider the image to possess that specific concept attribute and set the concept label to be *True*. We can obtain concept labels for all images $c = \{c_1, \ldots, c_M\}$, where $C_i \in \{0, 1\}^N$ is the concept label for the $i$-th sample. Finally, to ensure the truthfulness of concepts, we discard all concepts for which the similarity across all images is below 0.45. To achieve higher annotation accuracy, we only annotated 20% of the data for fine-tuning the model to adapt to different datasets. We sampled 10% of the pseudo-labels generated and compared them with expert annotations [11]. It is notable that in the case where the concept set and concept labels are given, we can directly skip Section 4.1 and 4.2.

### 4.3 Multi-dimensional Alignment

In Section 4.2, we get the concept labels for all input images. However, as we mentioned in the introduction, such representation might significantly degrade task accuracy [39, 67]. To overcome this issue, recently [67] propose using concept embeddings, which increase the task accuracy of concept-based models while weakening their interpretability. Motivated by this, we use these concept embeddings to increase the accuracy and leverage our concept label to enhance interpretability. In the following, we provide details.

**Concept Embeddings.** For the training data $X = \{(x_m, y_m)\}_{m=1}^M$, we use a backbone network (e.g., ResNet50) to extract features $\mathcal{F} = \{f(x_m)\}_{m=1}^M$. Then, for each feature, it passes through a concept encoder [67] to obtain feature embeddings $f_c(x_m)$ and concept embeddings $\hat{c}_m$. The specific process can be represented by the following expression:

$$f(x_m) = \Theta_b(x_m), f_c(x_m), \hat{C}_m = \Theta_c(f(x_m)) \text{ for m} \in [M],$$

where $\Theta_b$ and $\Theta_c$ represent the backbone and concept encoder, respectively.

To enhance the interpretability of concept embeddings, we utilize binary cross-entropy to optimize the accuracy of concept extraction by computing $\mathcal{L}_c$ based on $\hat{c} = \{\hat{c}_m\}_{m=1}^M$ and concept labels $c$ in Section 4.2:

$$\mathcal{L}_c = BCE(\hat{c}, c). \tag{1}$$

**Neural-Symbolic Layer**. Our next goal is to use concept embedding to learn concept rules for prediction, which is motivated by [3]. We aim to generate rules involving the utilization of two sets of feed-forward neural networks: $\Phi(\cdot)$ and $\Psi(\cdot)$. The output of $\Phi(\cdot)$ signifies the assigned role of each concept, determining whether it is positive or negative (such as "no LC" and "GO"). On the

other hand, the output of $\Psi(\cdot)$ represents the relevance of each concept, indicating whether it is useful within the context of the sample feature (such as "LDP"). The overall process can be divided into three distinct parts: (i) Learning the role (positive or negative) of each concept called *Concept Polarity*. For each prediction class $j$, there exists a neural network $\Phi_j(\cdot)$. This network takes each concept embedding as input and produces a soft indicator, a scalar value in the $[0, 1]$ range. This soft indicator represents the role of the concept within the formula; (ii) Learning the relevance of each concept called *Concept Relevance*. Similar to concept polarity, for each prediction class $j$, a neural network $\Psi_j(\cdot)$ is utilized; (iii) Output the logical reasoning rules of the concept and the contribution scores of concepts. For each class $j$, we combine the previous concept polarity vector $I_{o,j}$ and the concept relevance vector $I_{r,j}$ to obtain the logical inference output. This is achieved by the following expression (Details are in Appendix B):

$$\hat{y}_j = \wedge_{i=1}^{N}(\neg I_{o,i,j} \vee I_{r,i,j}) = \min_{i \in [N]}\{\max\{1 - I_{o,i,j}, I_{r,i,j}\}\}. \tag{2}$$

## 4.4 Final Objective

In this section, we will discuss how we derive the class of medical images and the process of network optimization. First, we have the loss $\mathcal{L}_c$ in (1) for enhancing the interpretability of concept embeddings. Also, as the concept embeddings are input into the neural-symbolic layer to output logical reasoning rules of the concept and prediction $\hat{y}_{neural,m}$ in (2) for $x_m$, we also have a loss between the predictions given by concept rule and ground truth, which corresponds to the interpretability of our neural-symbolic layers. In the context of binary classification tasks, we employ binary cross-entropy (BCE) as our loss function. For multi-class classification tasks, we use cross-entropy as the measure. Using binary classification as an example, we calculate the loss $\mathcal{L}_{neural}$ by comparing the output $\hat{y}$ from the neural-symbolic layer to the label $y$ as follows:

$$\mathcal{L}_{neural} = BCE(\hat{\boldsymbol{y}}_{neural}, \boldsymbol{y}), \tag{3}$$

**Classification Loss.** Note that as $\mathcal{L}_{neural}$ in (3) is purely dependent on the concept rules rather than feature embeddings, we still need a loss for final prediction performance. In a typical classification network, the process involves obtaining the feature $f(x_m)$ and passing it through a classification head to generate the classification results. What sets our approach apart is that we fuse the previously extracted $f_c(x_m)$ with the $f(x_m)$ using a fusion module as input to the classification ahead. This can be expressed using the following formula:

$$\tilde{y}_m = W_F \cdot \text{Concat}(f(x_m), f_c(x_m)),$$

Note that $W_F$ represents a fully connected neural network. For training our classification model, we use categorical cross-entropy loss, which is defined as follows:

$$\mathcal{L}_{task} = CE(\tilde{\boldsymbol{y}}, \boldsymbol{y}),$$

Formally, the overall loss function of our approach can be formulated as:

$$\mathcal{L} = \mathcal{L}_{task} + \lambda_1 \cdot \mathcal{L}_c + \lambda_2 \cdot \mathcal{L}_{neural},$$

where $\lambda_1, \lambda_2$ are hyperparameters for the trade-off between interpretability and accuracy.

## 5 Experiment

In this section, we introduce the experimental settings, present our superior performance, and showcase the interpretability of our network. Due to the space limit, additional experimental details and results are in the appendix C.

### 5.1 Experimental Setting

**Datasets.** We consider four benchmark medical datasets: COVID-CT [29] for CT images, DDI [10] for dermatology images, Chest X-Ray [14], and Fitzpatrick17k [15] for a dermatological dataset with skin colors.

**Baselines.** We compared our model with other state-of-the-art interpretable models, such as Label-free CBM[41] and DCR[3], to highlight the robustness of our interpretability capabilities. Furthermore, we conducted comparisons with black-box models, such as SSSD-COVID [51].

**Evaluation Metrics.** In this study, we concentrate on medical image classification. To comprehensively evaluate our classification model, we employ a range of key metrics, including binary accuracy, precision, recall, F1-Score, and AUC value. Specifically, we calculate each metric for positive and negative samples in each dataset, alongside utilizing AUC for further insight into classification performance.

**Experimental Setup.** We employed three different models as our backbones: ResNet-50 [19], VGG19 [48], and DenseNet169 [30]. To demonstrate the superiority of our method, we use the same backbones without any additional processing. For the concept encoder, we utilized the same structure as described in [35]. Before training, we obtained pseudo-labels for the concepts in each image using the automatic labeling process. Although previous work has shown that operations like super-resolution reconstruction can improve model accuracy on our dataset, to demonstrate the superiority of our method, we used the original images as our input. During training, we adopted the Adam optimizer with a learning rate of 5e-5 throughout the training stages. For hyperparameter selection, we set $\lambda_1 = \lambda_2 = 0.1$.

| Method | Backbone | COVID-CT | | DDI | | Chest X-Ray | | Fitzpatrick17k | | Interpretability |
| --- | --- | --- | --- | --- | --- | --- | --- | --- | --- | --- |
| | | Acc.(%) | F1(%) | Acc.(%) | F1(%) | Acc.(%) | F1(%) | Acc.(%) | F1(%) | |
| | ResNet50 | 81.36 | 81.67 | 77.27 | 72.77 | 75.64 | 71.72 | **80.79** | 80.79 | ✗ |
| | VGG19 | 79.60 | 79.88 | 76.52 | 70.12 | **81.41** | **77.56** | 75.37 | 75.37 | ✗ |
| Baseline | DenseNet169 | **85.59** | **85.59** | 78.03 | 69.51 | 69.55 | 61.66 | 76.85 | 76.83 | ✗ |
| | SSSD-COVID | 81.76 | 80.00 | - | - | - | - | - | - | ✗ |
| | Label Free CBM | 69.49 | 69.21 | 70.34 | 69.21 | 71.21 | 70.84 | 75.24 | 75.41 | ✓ |
| | DCR | 55.93 | 51.41 | 76.52 | 65.32 | 62.02 | 41.33 | 68.05 | 66.12 | ✓ |
| | ResNet50 | 84.75 | 84.75 | **81.82** | **76.33** | 78.37 | 74.42 | **82.76** | **83.03** | ✓ |
| Ours | VGG19 | 83.05 | 84.37 | **82.58** | **78.07** | **88.30** | **88.16** | 77.34 | 77.53 | ✓ |
| | DenseNet169 | **86.44** | **87.15** | 79.55 | 69.79 | 73.88 | 65.70 | 80.79 | **81.11** | ✓ |

Table 1: Utility results. We conducted ten repeated experiments and calculated the average results for each metric. Red and blue indicate the best and the second-best result. Our approach outperforms the baseline backbone models while also providing interpretability. Our model exhibits significantly higher accuracy compared to other state-of-the-art interpretable models.

## 5.2 Model Utility Analysis

**Med-MICN delivers superior performance.** In Table 1, our method achieved different improvements on various backbones for the COVID-CT dataset. Taking Acc as an example, it increased by 3.39% with ResNet50 and by 3.45% with VGG compared to backbones. Compared to SSSD-COVID, our method outperforms in terms of Acc and F1, with improvements of 4.68% and 7.15%, demonstrating the superiority of our method in enhancing model accuracy. Additionally, our method possesses interpretability, which is not achievable by SSSD-COVID. On the other hand, our method achieved significant improvements on different backbones for the DDI dataset. For instance, Acc increased by 6.01% with VGG. Despite the significant differences between the two datasets in terms of modality, our method demonstrated significant effects on both datasets, indicating its good generalizability. Details are shown in Table 4, 5, 6, and 7 in Appendix.

Meanwhile, when compared to existing well-performing interpretable models, our approach demonstrates significant advantages in terms of accuracy and other metrics across different datasets. This indicates that our joint prediction of image categories using both concept and image feature spaces outperforms predictions based solely on a limited concept space.

## 5.3 Model Interpretability Analysis

**Explanation across multiple dimensions.** In our approach, we discover and generate concept reasoning rules based on the neural-symbolic layer. Analyzing these rules enhances the interpretability of our network. In addition, our approach derives concept prediction scores through the concept encoder, and during evaluation, it also produces saliency maps. Through multi-dimensional explanations, we can observe the basis for the model decision from different perspectives. In concept score prediction, we can observe how the model maps data to concept dimensions, allowing doctors to observe and correct concept results, thereby rectifying prediction errors caused by incorrect concept predictions. In concept reasoning rules, we can deduce the model classification criteria, further explaining the model decisions. Additionally, during the evaluation process, we can generate saliency maps for the

| Dataset | Ablation Setting | | Metrics | | | | | |
| --- | $\mathcal{L}_c$ | $\mathcal{L}_{neural}$ | ACC.(%) | Precision(%) | Recall(%) | F1(%) | AUC.(%) | Interpretability |
| --- | --- | --- | --- | --- | --- | --- | --- | --- |
| COVID-CT | | | 82.20 | 82.92 | 82.21 | 82.55 | 82.64 | |
| | ✓ | | 83.05 | 83.62 | 83.16 | 83.01 | 83.16 | |
| | | ✓ | 81.36 | 82.11 | 81.38 | 81.70 | 81.81 | |
| | ✓ | ✓ | **84.75** | **84.77** | **84.88** | **84.75** | **84.77** | ✓ |
| DDI | | | 78.03 | 74.97 | 66.88 | 69.24 | 67.41 | |
| | ✓ | | 79.55 | 75.36 | 71.47 | 72.73 | 71.20 | |
| | | ✓ | 78.79 | 76.38 | 66.29 | 68.69 | 67.64 | |
| | ✓ | ✓ | **81.82** | **76.56** | **76.17** | **76.33** | **76.12** | ✓ |
| Chest X-Ray | | | 68.59 | 69.63 | 61.11 | 61.02 | 62.05 | |
| | ✓ | | 72.28 | 77.63 | 64.15 | 63.72 | 64.15 | |
| | | ✓ | 70.03 | 73.83 | 61.84 | 61.25 | 62.39 | |
| | ✓ | ✓ | **78.37** | **80.38** | **73.12** | **74.42** | **73.12** | ✓ |
| Fitzpatrick17k | | | 78.33 | 79.50 | 78.32 | 78.91 | 79.06 | |
| | ✓ | | 79.80 | 80.60 | 79.81 | 80.20 | 80.31 | |
| | | ✓ | 80.79 | 81.28 | 80.82 | 81.28 | 81.07 | |
| | ✓ | ✓ | **82.76** | **82.84** | **83.23** | **83.03** | **82.99** | ✓ |

Table 2: Experimental results from ablation studies on each loss function demonstrate that each loss function is indispensable for both accuracy and interpretability.

data, providing an intuitive understanding of how pixels in the image influence the image classification result aligned with semantic concepts. Compared to traditional concept-based models, our model interpretation offers advantages in richness and accuracy. Additional visualization examples can be found in Appendix D.2.

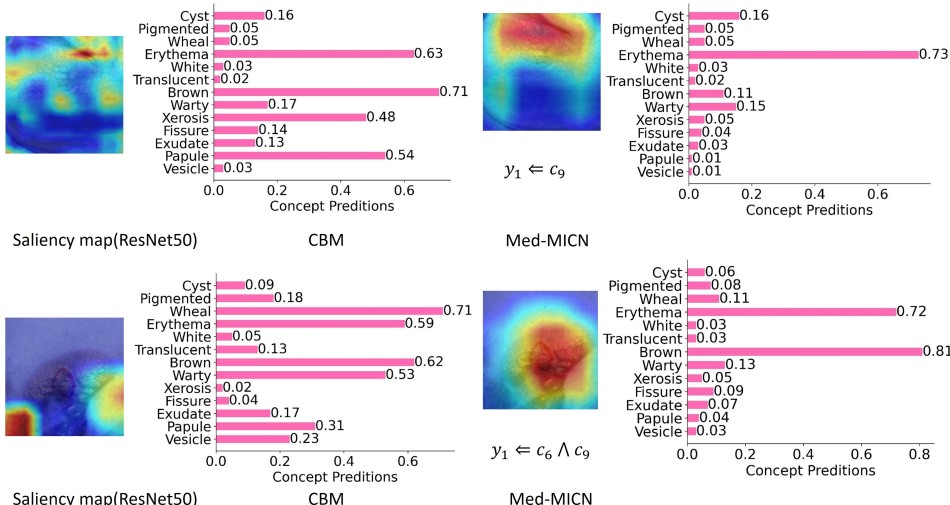

Figure 4: Comparison of single-dimensional and multi-dimensional interpretability methods.

As shown in Figure 4, it is evident that relying solely on single-dimensional interpretable strategies, such as saliency maps or concept embedding enhancements, does not furnish adequate interpretability to effectively address the problem. However, by integrating multi-dimensional strategies, the model can align the information of each dimension, thus obtaining more comprehensive interpretable information and ultimately yielding more correct prediction results. Specifically, when feature extraction is solely reliant on saliency maps, obtaining accurate attention in the feature region often proves challenging, and conceptual information tends to be unstable when supplemented solely by concepts. In contrast, our proposed multi-dimensional interpretable strategy transcends dependence on a single interpretable strategy, opting instead for a more generalized multi-dimensional augmentation approach. This approach enables the model to complement the single-dimensional methods and achieve heightened accuracy.

### 5.4 Ablation Study

The ablation experiments presented in Table 2, conducted with Resnet50 as the backbone, reveal significant contributions from both $\mathcal{L}_c$ and $\mathcal{L}_{neural}$ to the classification result. To illustrate, considering the comprehensive index AUC in the DDI dataset, utilizing only $\mathcal{L}_c$ yields a 3.79% improvement, while relying solely on $\mathcal{L}_{neural}$ does not notably enhance performance. However, employing both

simultaneously leads to an 8.71% improvement. This observation underscores the complementary nature of concept and neural logic rules in enhancing model performance. Besides, additional ablation studies investigating the effect of concept filters and comparing VLMs labeling methods with other Med-CLIP approaches are provided in Appendix D.3. Additionally, we conducted a sensitivity analysis for both baselines and Med-MICN on the DDI dataset, as depicted in Figure 14 in the appendix, showcasing the robustness of our model against perturbations. Furthermore, computational cost analysis (presented in Table 9 in the appendix) was conducted. Experimental findings indicate that Med-MICN incurs minimal computational cost compared to the baseline model while achieving improvements in both accuracy and interpretability.

## 6 Conclusion

This paper proposes a novel end-to-end interpretable concept-based model called Med-MICN. Combining medical image classification, neural symbolic solving, and concept semantics, Med-MICN achieves superior accuracy and multi-dimensional interpretability. Our comprehensive experiments demonstrate consistent enhancements over other baselines, highlighting its potential as a generalized and faithful interpretation model for medical images.

## Acknowledgments

Di Wang and Lijie Hu are supported in part by the funding BAS/1/1689-01-01, URF/1/4663-01-01, REI/1/5232-01-01, REI/1/5332-01-01, and URF/1/5508-01-01 from KAUST, and funding from KAUST - Center of Excellence for Generative AI, under award number 5940. Di Wang and Lijie Hu are also supported by the funding of the SDAIA-KAUST Center of Excellence in Data Science and Artificial Intelligence (SDAIA-KAUST AI).

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

# A  Concept Set Generation and Filtering

## A.1  Concept Generation with GPT-4V

In this section, we will explain our process for generating concept sets using GPT-4V. Specifically, we ask GPT-4V the following:

> **[User]:**
> - List the useful visual attributes to recognize {class} in a {type} in this {image}:
> - List the things most commonly seen around a {class} in this {image}:
> - List in bullet points with no extra context.

Note that the image here refers to the images we random sample from the dataset, and type represents the modality of medical image (e.g. X-rays, CT). Full prompts and example outputs are illustrated in the appendix D.2. Combining the concepts received from different classes and images gives us a large, somewhat noisy set of initial concepts, which we further improve by filtering. We found using GPT-4V to generate initial concepts to perform better than using GPT-3, which was used in Label-free CBM [41]. We initiate the procedure with the following conversation:

> **[User]:** List the useful visual attributes to recognize COVID in a CT in this image.
> **[GPT-4V]:** When assessing CT images for signs of COVID-19, radiologists often look for certain visual attributes that can suggest the presence of the disease. Here are some of the key visual attributes they may look for:
> 1. **Ground-glass opacities (GGOs)**: These are hazy areas that do not obscure the underlying lung structures. They are called "ground-glass" because they resemble the appearance of frosted glass.
> 2. **Consolidation**: ...

In addition, we can proceed with the following question:

> **[User]:** List the things most commonly seen around COVID in this image.
> **[GPT-4V]:** Based on the image you have provided and considering the common radiological findings associated with COVID-19, here are the things that might be commonly seen:
> 1. Possible consolidation areas.
> 2. Absence of significant pleural effusions.
> 3. ......

Then, we can use the following inquiry to structure the output from GPT-4V:

> **[User]:** List in bullet points with no extra context.
> **[GPT-4V]:**
> - *Ground-Glass Opacities (GGOs)*
> - *Bilateral and Multifocal Lesions*
> - *Crazy-Paving Pattern*
> - *Reverse Halo or Atoll Sign*
> - *Absence of Pleural Effusion*
> - *Septal Thickening*
> - *Rare Lymphadenopathy*
> - *Vascular Enlargement*
> - *Traction Bronchiectasis*

After obtaining the initial concept set following the filtering process, our rules and concepts on COVID-CT and DDI datasets are as follows (see Table 3).

| Rules | COVID-CT | DDI |
|-------|----------|-----|
| $c_0$ | Peripheral ground-glass opacities | Vesicle |
| $c_1$ | Bilateral involvement | Papule |
| $c_2$ | Multilobar distribution | Exudate |
| $c_3$ | Crazy-paving pattern | Fissure |
| $c_4$ | Absence of lobar consolidation | Xerosis |
| $c_5$ | Localized or diffuse persentation | Warty |
| $c_6$ | lncreased density in the lung | Brown |
| $c_7$ | Ground-glass appearance | Translucent |
| $c_8$ | - | White |
| $c_9$ | - | Erythema |
| $c_{10}$ | - | Wheal |
| $c_{11}$ | - | Pigmented |
| $c_{12}$ | - | Cyst |

Table 3: We have recorded the concept set generated through GPT-4V and subsequently cleaned it through filtering. This concept set will be used in our training process.

## A.2   Concept Set Filtering

After concept set generation, a concept set with some noise can be obtained. The following filters are set to enhance the quality of the concept :

1. **The length of concept**: To keep concepts simple and avoid unnecessary complication, We remove concepts longer than 30 characters in length.

2. **Similarity**: We measure this with cosine similarity in a text embedding space. We removed concepts too similar to classes or each other. The former conflicts with our interpretability goals, while the latter leads to redundancy in concepts. We set the thresholds for these two filters at 0.85 and 0.9, respectively, ensuring that their similarities are below our threshold.

3. **Remove concepts we cannot project accurately**: Remove neurons that are not interpretable from the BioViL [5]. This step is actually described in section 4.2.

# B   Neural-symbolic Layer

We give the details with examples for neural-symbolic layer [3].

**Concept Polarity.**    For each prediction class $j$, there exists a neural network $\Phi_j(\cdot)$. This network takes each concept embedding as input and produces a soft indicator, a scalar value in the $[0, 1]$ range. This soft indicator represents the role of the concept within the formula. As an illustration, consider a specific concept like "Crazy-paving pattern". If its value after passing through $\Phi_j(\cdot)$ is 0.8, it indicates that the "Crazy-paving pattern" has a positive role with a score of 0.8 for class $j$. We use the notation $I_{o,i,j}$ to represent the soft indicator for concept $c_i$.

**Concept Relevance.**    For each prediction class $j$, a neural network $\Psi_j(\cdot)$ is utilized. This network takes each concept embedding as input and produces a soft indicator, a scalar value within the range of $[0, 1]$, representing the relevance score within the formula. To illustrate, let us consider a specific concept such as "Multilobar distribution". If its value after passing through $\Psi_j(\cdot)$ is 0.2, it implies that the relevance score of "Multilobar distribution" in the inference of class $j$ is 0.2. We denote the soft indicator for concept $c_i$ as $I_{r,i,j}$.

**Prediction via Concept Rules.**    Finally, for each class $j$, we combine the previous concept polarity vector $I_{o,j}$ and the concept relevance vector $I_{r,j}$ to obtain the logical inference output. This is

achieved by the following expression:

$$\hat{y}_j = \wedge_{i=1}^{N} (\neg I_{o,i,j} \vee I_{r,i,j})$$
$$= \min_{i \in [N]} \{\max\{1 - I_{o,i,j}, I_{r,i,j}\}\},$$

Intuitively, the aforementioned concept reasoning rules allow us to determine the relevance of each concept and whether a concept has a positive or negative role in predicting the label as class $j$. Intuitively $\neg I_{o,i,j} \vee I_{r,i,j}$ means if it is irrelevant, then we set it to 1, and if it is relevant, then we set it to be the relevance score. It is strange at first glance. However, since finally we will take the conjunction (or minimum) for all concepts to get $\hat{y}_j$, so we will filter the $i$-th concept if $\neg I_{o,i,j} \vee I_{r,i,j} = 1$, i.e., if it is irrelevant. Thus, we take the neg here. Since we need to consider each concept for the final prediction, we finally use $\wedge$ for all concepts.

## C    More Experimental Setup

### C.1    Training Setting

Our model exhibits remarkable efficiency in its training process. We utilized only a single GeForce RTX 4090 GPU, and the training duration did not exceed half an hour. We configured the model to run for 100 epochs with a learning rate set at 5e-5. Additionally, all images were resized to a uniform dimension of (256, 256).

### C.2    Datasets

**COVID-CT.**    The COVID-CT dataset was obtained from [29] and comprises 746 CT images, consisting of two classes (349 COVID and 397 NonCOVID). We divided this dataset into a training set and a test set with an 8:2 ratio, and the data were split accordingly.

**DDI.**    DDI [10] is a dataset comprising diverse dermatology images designed to assess the model's ability to correctly identify skin diseases. It consists of a total of 656 images, including 485 benign lesion images and 171 malignant lesion images. These images are divided into training and test sets, with an 80% and 20% split, respectively.

**Chest X-Ray.**    The Chest X-Ray [14] dataset comprises 2D chest X-ray images of both healthy and infected populations. It aims to support researchers in developing artificial intelligence models capable of distinguishing between chest X-ray images of healthy individuals and those with infections. The dataset includes 5933 images, divided into 5309 training images and 624 testing images.

**Fitzpatrick17k.**    Fitzpatrick17k [15] dataset is a dermatological dataset that includes a wide range of skin colors. In order to better compare the model performance, we filtered the malignant and nonmalignant classes in 3230 images that have been relabeled by SkinCon [11]. And we divided the training and test set according to an 8:2 ratio.

### C.3    Baseline Models

**SSSD-COVID.**    SSSD-COVID incorporates a Masked Autoencoder (MAE) for direct pre-training and fine-tuning on a small-scale target dataset. It leverages self-supervised learning and self-distillation techniques for COVID-19 medical image classification, achieving performance levels surpassing many baseline models. Our method is trained exclusively on the COVID-CT dataset without considering the effects of introducing knowledge from other datasets. Notably, our approach outperforms SSSD-COVID in terms of performance. Furthermore, SSSD-COVID falls under the category of black-box models, indicating that our method, to some extent, overcomes the accuracy degradation issue introduced by the incorporation of concepts.

**Label-free CBM.**    Label-free CBM is a fully automated and scalable method for generating concept bottleneck models. It has demonstrated outstanding performance on datasets like ImageNet. In our comparisons, our model outperforms this model significantly in terms of accuracy. Regarding interpretability, our model not only possesses the same level of interpretability as this model but also

provides explanations in multiple dimensions, such as concept reasoning and saliency maps. This helps doctors gain a more diverse and precise understanding of the model's decision-making process when using our model.

**DCR.** Deep Concept Reasoner (DCR), a concept-based model combining neural-symbolic methods, has demonstrated promising interpretability performance on simple datasets such as XOR, Trigonometry, and MNIST-Addition. However, its performance tends to degrade on more complex or challenging datasets.

## C.4 Definition of Metrics

Accuracy is the ratio of the number of correctly categorized samples to the total number of samples:

$$\text{Acc} = \frac{\text{TP} + \text{TN}}{\text{TP} + \text{TN} + \text{FP} + \text{FN}}.$$

Where TP denotes true positive, TN denotes true negative, FP represents false negative, and FN represents false negative.

The precision is the proportion of all samples classified as positive categories that are actually positive categories:

$$\text{Precision} = \frac{\text{TP}}{\text{TP} + \text{FP}}.$$

Recall is the proportion of samples that are correctly categorized as positive out of all samples that are actually positive categories:

$$\text{Recall} = \frac{\text{TP}}{\text{TP} + \text{FN}}.$$

The F1 score is the reconciled mean of precision and recall:

$$\text{F1} = \frac{2 \times \text{Precision} \times \text{Recall}}{\text{Precision} + \text{Recall}}.$$

AUC denotes the area under the ROC curve, which is a curve with True Positive Rate (Recall Rate) as the vertical axis and False Positive Rate (False Positive Rate) as the horizontal axis.

# D More Experimental Results

## D.1 Utility Evaluation

We provide our detailed utility evaluation for four datasets in Table 4, 5, 6, and 7.

| Method | Backbone | Acc.(%) | Precision(%) | Recall(%) | F1(%) | AUC.(%) | Interpretability |
|---|---|---|---|---|---|---|---|
| **Baseline** | ResNet50 | 81.36 | 82.28 | 81.44 | 81.67 | 81.85 | ✗ |
| | VGG19 | 79.60 | 81.82 | 78.93 | 79.88 | 80.26 | ✗ |
| | DenseNet169 | **85.59** | 85.60 | **85.60** | **85.59** | 85.60 | ✗ |
| | SSSD-COVID | 81.76 | 81.82 | 78.26 | 80.00 | **88.21** | ✗ |
| | Label Free CBM | 69.49 | 68.62 | 69.82 | 69.21 | 64.84 | ✓ |
| | DCR | 55.93 | 58.38 | 55.43 | 51.41 | 55.43 | ✓ |
| **Ours** | ResNet50 | 84.75 | 84.77 | 84.88 | 84.75 | 84.77 | ✓ |
| | VGG19 | 83.05 | **86.74** | 82.93 | 84.37 | 84.26 | ✓ |
| | DenseNet169 | **86.44** | **87.27** | **86.41** | **87.15** | **87.92** | ✓ |

Table 4: **Results for COVID-CT.** We conducted ten repeated experiments and calculated the average results for each metric. Red and blue indicate the best and the second-best result. Our approach outperforms the baseline backbone models while also providing interpretability. When compared to other state-of-the-art interpretable models, our model exhibits significantly higher accuracy.

| Method | Backbone | Acc.(%) | Precision(%) | Recall(%) | F1(%) | AUC.(%) | Interpretability |
|--------|----------|---------|--------------|-----------|-------|---------|------------------|
| **Baseline** | ResNet50 | 77.27 | 72.37 | 73.19 | 72.77 | 72.51 | ✗ |
| | VGG19 | 76.52 | 72.92 | 68.54 | 70.12 | 68.80 | ✗ |
| | DenseNet169 | 78.03 | 74.37 | 67.41 | 69.51 | 68.76 | ✗ |
| | Label Free CBM | 70.34 | 68.62 | 69.82 | 69.21 | 69.49 | ✓ |
| | DCR | 76.52 | 71.79 | 63.88 | 65.32 | 63.88 | ✓ |
| **Ours** | ResNet50 | 81.82 | 76.56 | 76.17 | 76.33 | 76.12 | ✓ |
| | VGG19 | 82.58 | 81.59 | 76.05 | 78.07 | 75.63 | ✓ |
| | DenseNet169 | 79.55 | 77.68 | 67.64 | 69.79 | 67.64 | ✓ |

Table 5: **Results for DDI.** We conducted ten repeated experiments and calculated the average results for each metric. Red and blue indicate the best and the second-best result. Our approach outperforms the baseline backbone models while also providing interpretability. When compared to other state-of-the-art interpretable models, our model exhibits significantly higher accuracy.

| Method | Backbone | Acc.(%) | Precision(%) | Recall(%) | F1(%) | AUC.(%) | Interpretability |
|--------|----------|---------|--------------|-----------|-------|---------|------------------|
| **Baseline** | ResNet50 | 75.64 | 75.01 | 70.77 | 71.72 | 70.88 | ✗ |
| | VGG19 | 81.41 | 88.56 | 75.51 | 77.56 | 75.94 | ✗ |
| | DenseNet169 | 69.55 | 70.37 | 62.05 | 61.66 | 62.12 | ✗ |
| | Label Free CBM | 71.21 | 71.89 | 71.45 | 70.84 | 74.12 | ✓ |
| | DCR | 62.02 | 66.25 | 51.50 | 41.33 | 50.56 | ✓ |
| **Ours** | ResNet50 | 78.37 | 80.38 | 73.12 | 74.42 | 73.12 | ✓ |
| | VGG19 | 88.30 | 92.59 | 85.43 | 88.16 | 87.09 | ✓ |
| | DenseNet169 | 73.88 | 81.24 | 65.85 | 65.70 | 66.28 | ✓ |

Table 6: **Results for Chest X-Ray.** We conducted ten repeated experiments and calculated the average results for each metric. Red and blue indicate the best and the second-best result. Our approach outperforms the baseline backbone models while also providing interpretability. When compared to other state-of-the-art interpretable models, our model exhibits significantly higher accuracy.

| Method | Backbone | Acc.(%) | Precision(%) | Recall(%) | F1(%) | AUC.(%) | Interpretability |
|--------|----------|---------|--------------|-----------|-------|---------|------------------|
| **Baseline** | ResNet50 | 80.79 | 80.81 | 80.81 | 80.79 | 80.81 | ✗ |
| | VGG19 | 75.37 | 75.40 | 75.34 | 75.37 | 75.39 | ✗ |
| | DenseNet169 | 76.85 | 77.05 | 76.91 | 76.83 | 76.91 | ✗ |
| | Label Free CBM | 75.24 | 75.15 | 74.92 | 75.41 | 75.02 | ✓ |
| | DCR | 68.05 | 67.55 | 65.33 | 66.12 | 67.01 | ✓ |
| **Ours** | ResNet50 | 82.76 | 82.84 | 83.23 | 83.03 | 82.99 | ✓ |
| | VGG19 | 77.34 | 77.72 | 77.33 | 77.53 | 77.58 | ✓ |
| | DenseNet169 | 80.79 | 82.12 | 80.89 | 81.11 | 81.38 | ✓ |

Table 7: **Results for Fitzpatrick17k.** We conducted ten repeated experiments and calculated the average results for each metric. Red and blue indicate the best and the second-best result. Our approach outperforms the baseline backbone models while also providing interpretability. When compared to other state-of-the-art interpretable models, our model exhibits significantly higher accuracy.

### D.2 Visualization of Concept Predictions

More samples of instance-level predictions for COVID-CT and DDI datasets are visualized in Figure 5, 6, 7, 8, 9, 10, 11, and 12. We also append the corresponding rules to it, which can reflect the multi-dimensional interpretation better. Taking the COVID-CT sample as an example, it can be found that the model predicts the concept score prediction accurately, and for the COVID samples, the correlation of the first three concepts is greater, which leads to a higher prediction score of their concepts, thus generating real concept rules, and then assists the model judgment. This can also be reflected in the saliency map of the model in the inference stage, where the phenomena described by the relevant concepts can get higher attention in the saliency map, thus further reflecting the multidimensional interpretation of Med-MICN.

### D.3 Ablation Study: Effect of Concept Filters

In this section, we will discuss how each step in our proposed concept filtering affects the results of our method. In general, our utilization of filters has two main goals: First, improving the interpretability of our models. Second, improving computational efficiency and complexity by reducing the number of concepts.

Although our initial goal was not to improve model accuracy (a model with more concepts is generally larger and more powerful [41]), the more precise concepts after filtering make the concatenated features more effective for classification and slightly improve the accuracy. To evaluate the impact of each filter, we trained our models separately on COVID-CT and DDI while removing one filter at a time and one without using any filter at all. The results are shown in the Table 8. From Table 8, it is noticeable that our model's accuracy does not exhibit high sensitivity to the choice of filters. On the COVID-CT dataset, the accuracy of our model remains largely unaffected by the choice of filters. Furthermore, employing filters on the DDI dataset results in improved model accuracy. This phenomenon arises due to the relatively sparse nature of concepts within the DDI dataset images, where increasing the number of concepts did not yield superior solutions.

| Filters | COVID-CT | | DDI | |
|---|---|---|---|---|
| | Accuracy(%) | #concept | Accuracy(%) | #concept |
| All filters | 86.44 | 8 | 82.58 | 13 |
| No length filter | 86.32 | 9 | 82.53 | 16 |
| No similarity filter | 86.43 | 22 | 82.46 | 26 |
| No projection filter | 86.21 | 11 | 81.68 | 15 |
| No filters at all | 86.19 | 34 | 81.60 | 49 |

Table 8: Effect of our individual concept filters on the final accuracy and number of concepts utilized by our models.

### D.4 Sensitivity Analysis

We also performed a sensitivity analysis for baselines and Med-MICN in the DDI dataset. we set various attacks under $\delta \in 4/255, 6/255, 8/255, 10/255, 12/255$ and attack radius $\rho_a \in \{0, 2/255, 4/255, 6/255, 8/255, 10/255\}$. in Figure 14, the results show that our model consists of stronger robustness against perturbation. This is evidenced by the marginal decrease in test accuracy as the attack radius increases. The model's detection performance fluctuates slightly at $\rho_a$ of 4/255, with this variability diminishing as the perturbation level rises, underscoring the robustness of our model against significant disturbances.

### D.5 Computational Cost

We conducted a computational cost analysis for Med-MICN and the baseline models. By inputting a random tensor of size (1, 3, 244, 224) into the model and computing the FLOPs and parameters, the results are presented in Table 9. Experimental evidence demonstrates that Med-MICN incurs only negligible computational cost compared to the baseline models while it achieves improvements in accuracy and interpretability.

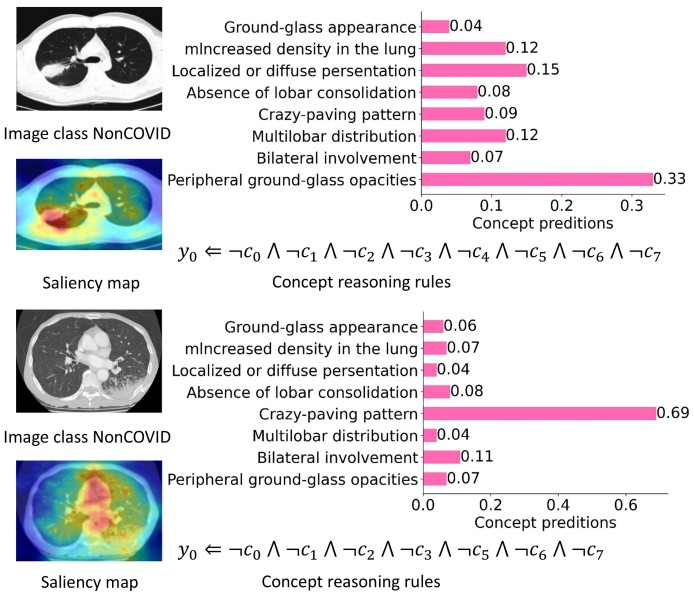

Figure 5: Samples classified as NonCOVID.

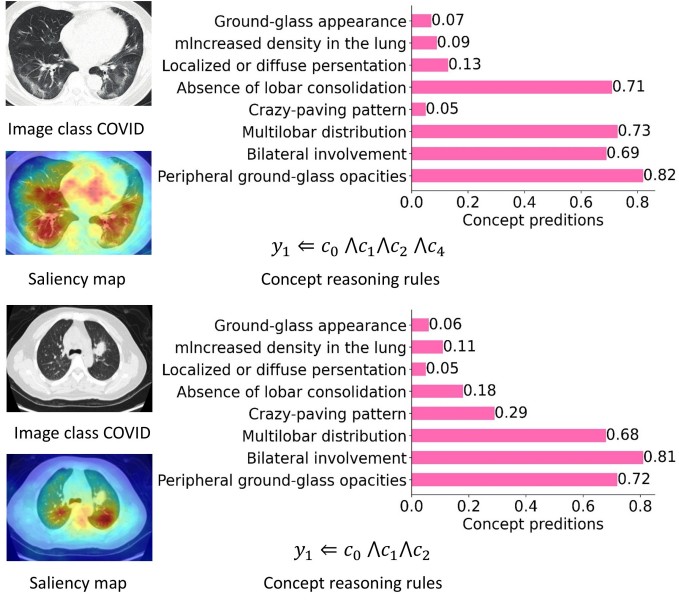

Figure 6: Samples classified as COVID.

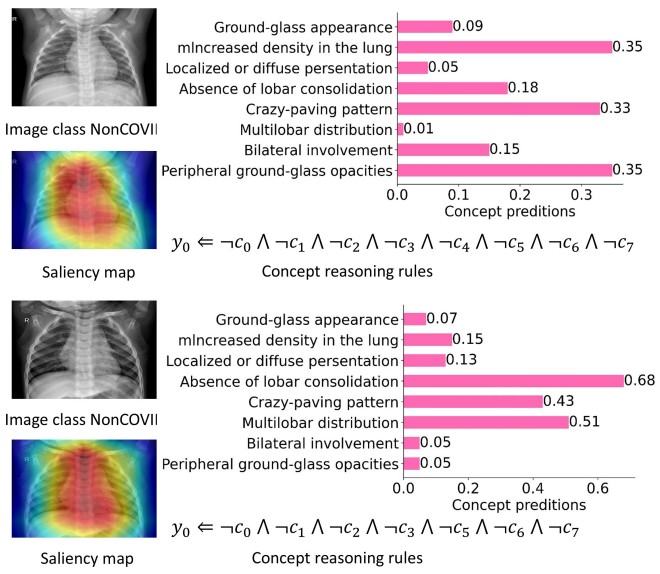

Figure 7: Samples classified as NonCOVID.

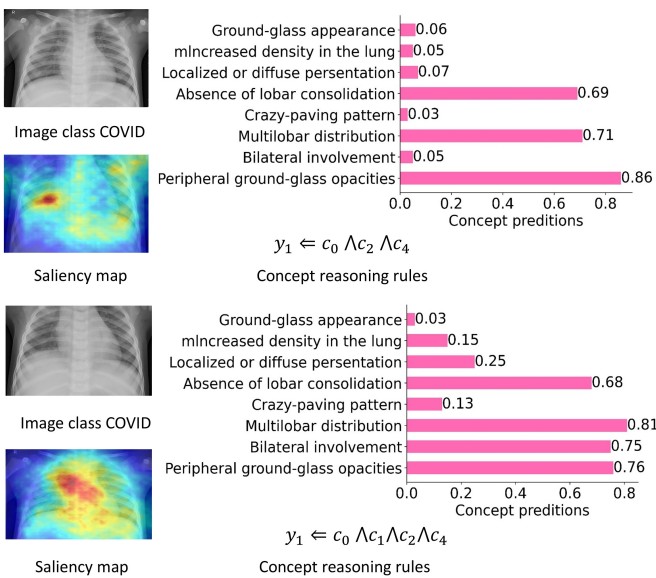

Figure 8: Samples classified as COVID.

| Method | Flops(M) | Params(M) |
|---|---|---|
| ResNet50 | 4133.74 | 25.56 |
| VGG16 | 15470.31 | 138.36 |
| DenseNet169 | 3436.12 | 14.15 |
| Med-ICNS(ResNet50) | 4134.79 | 26.60 |
| Med-ICNS(VGG16) | 15471.36 | 139.40 |
| Med-ICNS(DenseNet169) | 3436.32 | 14.35 |

Table 9: Computational costs. Results indicate that Med-MICN incurs minimal computational cost compared to the baseline model while achieving improvements in both accuracy and interpretability.

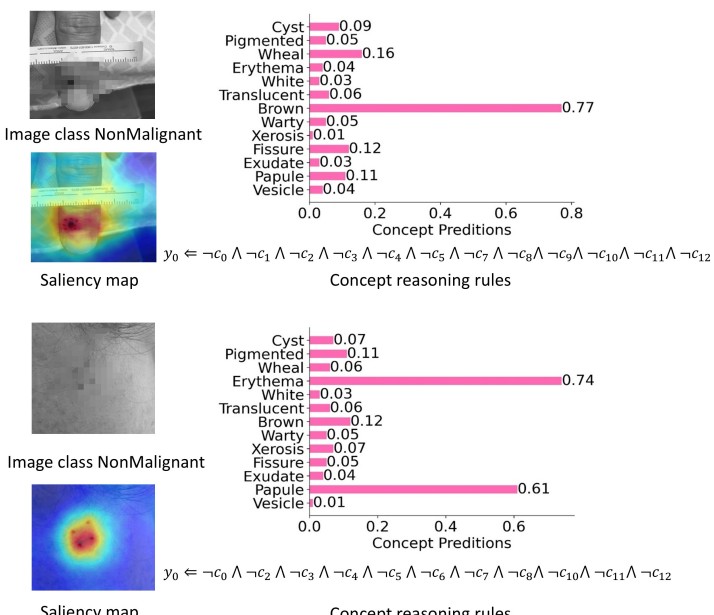

Figure 9: Samples classified as NonMalignant.

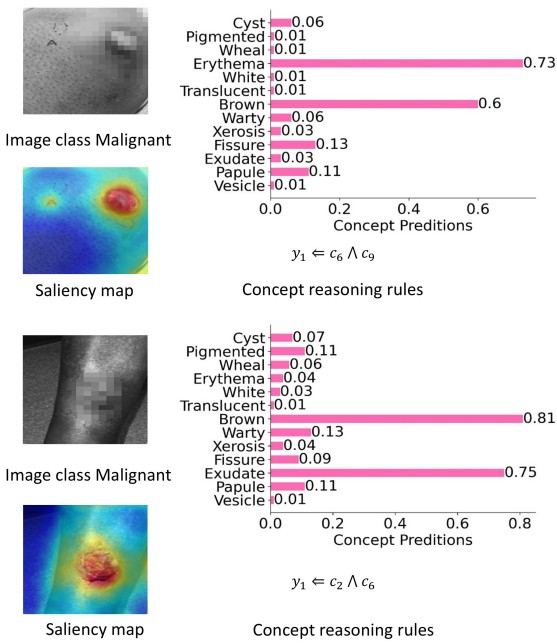

Figure 10: Samples classified as Malignant.

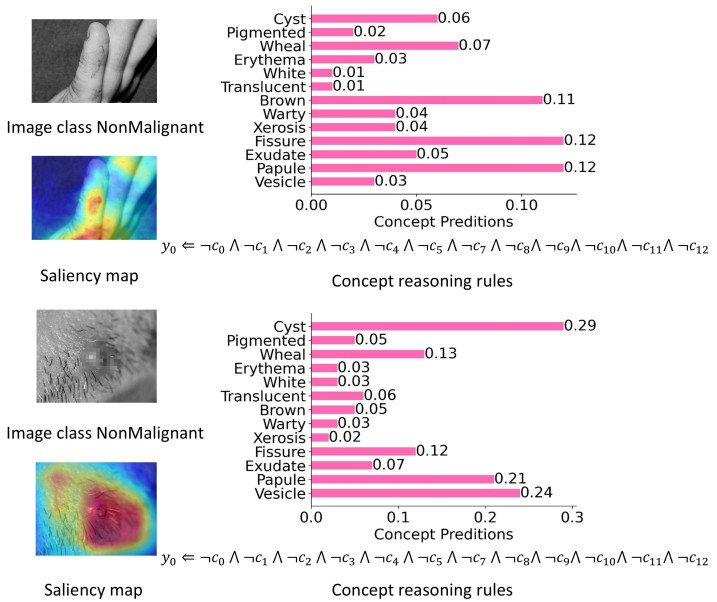

Figure 11: Samples classified as NonMalignant.

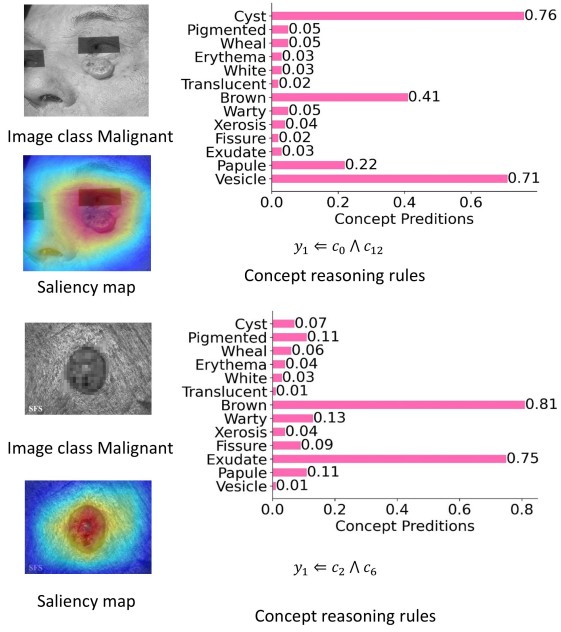

Figure 12: Samples classified as Malignant.

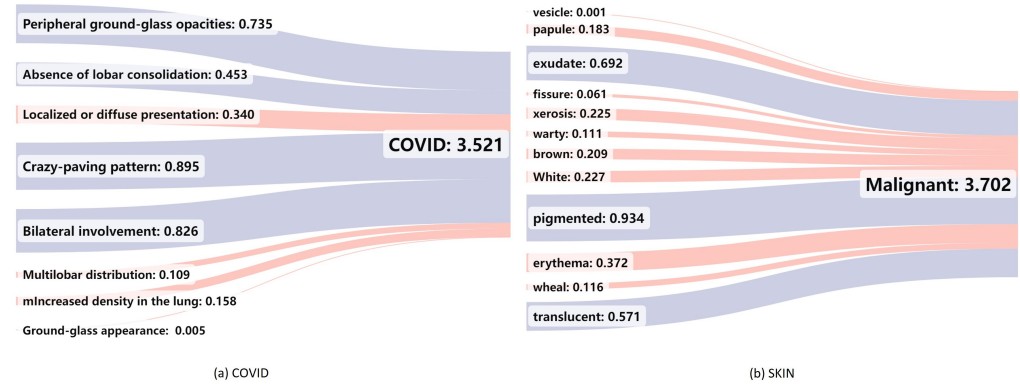

(a) COVID          (b) SKIN

Figure 13: Interpretation of learned linear weights for COVID-CT (left) and DDI (right) dataset. The model can discover concepts that are crucial for classification.

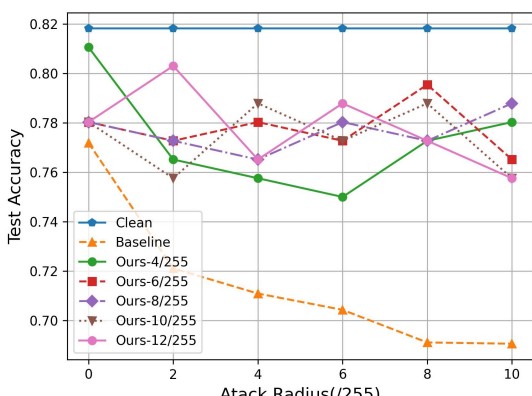

Figure 14: Sensitivity analysis.

# E  Limitation

In the case of the "neural symbolic" method, additional acceleration techniques may be required when dealing with large sample sizes. However, it is worth noting that medical datasets tend to be relatively small.

