# OpenReview forum: "Towards Multi-dimensional Explanation Alignment for Medical Classification"
_NeurIPS.cc/2024/Conference — NeurIPS 2024 poster_

### Official Review · Reviewer_gQUT · 2024-07-08

**Soundness:** 3
**Presentation:** 4
**Contribution:** 3
**Rating:** 8
**Confidence:** 5

**Summary:**

This work introduces a novel end-to-end concept-based framework called Med-MICN, which is quite inspiring and important for the next XAI era due to its multi-dimensional, powerful interpretability, and efficient performance. Furthermore, they propose an automated process for efficiently and accurately obtaining concept labels, which are costly to acquire in reality.

**Strengths:**

1. This paper introduces a novel attempt that creatively integrates medical image classification, neural symbolic solving, and concept semantics. This integration enables multidimensional interpretability, making it more comprehensive and powerful compared to previous interpretable models.

2. Med-MICN demonstrates superior accuracy and interpretability across datasets from different modalities.

3. The ample experiments are convincing, as the authors conducted experiments on multiple datasets and baselines to validate the model's interpretability and accuracy, while also providing rich and impressive visualization results.

**Weaknesses:**

1. The caption of the picture is not detailed enough, as shown in figure 3.

2. The author considers interpretability on the basis of concept embedding, neural-sybolic reasoning and saliency map. The author lacks the reason why it is necessary to analyze the interpretation from these aspects and whether there are other more dimensions.

**Questions:**

See weeknesses.

---

> ### Author Rebuttal · Authors · 2024-08-07
>
> We appreciate the reviewers' recognition of our work and your constructive comments. Please find below our detailed responses to the queries you have raised. We hope you could consider increasing our overall score if our response addresses your concerns :)
>
> > W1: "The caption of the picture is not detailed enough, as shown in figure 3."
>
> **Response**: Thank you for your suggestions. We have provided additional details for each figure caption as follows:
>
> - Figure 2: (a) module, output rich dimensional interpretable conceptual information for the specified disease through the multimodal model and convert the conceptual information into text vectors through the text embedding module; (b) module, access the image to the image embedder to get the image features, and then match with the conceptual textual information to get the relevant attention region. Then, we get the influence score of the relevant region information through pooling, and finally send it to the filter to sieve out the concept information with weak relevance to get the disease concept of image information.
> - Figure 3: Overview of the Med-MICN Framework. The Med-MICN framework consists of four primary modules: (1) **Feature Extraction Module**: In the initial step, image features are extracted using a backbone network to obtain pixel-level features. (2) **Concept Embedding Module**: The extracted features are fed into the concept embedding module. This module outputs concept embeddings while passing through a category classification linkage layer to obtain predicted category information. (3) **Concept Semantic Alignment**: Concurrently, a Vision-Language Model (VLM) is used to annotate the image features, generating concept category annotations aligned with the predicted categories. (4) **Neural Symbolic Layer**: After obtaining the concept embeddings, they are input into the Neural Symbolic layer to derive conceptual rules.  Finally, the concept embeddings obtained from module (2) are concatenated with the original image embeddings and fed into the final category prediction layer to produce the ultimate prediction results.
> - Figure 4: A full comparison of our method with current saliency and CBM methods in terms of interpretable dimensions and effectiveness is shown in Fig. 4. CBM class methods lack interpretation of saliency map and rules dimensions. saliency map class methods lack interpretation of concept and rules dimensions. For this medical classification task, single dimensions such as saliency map methods do not accurately capture the features of disease images, and CBM-type methods for concept extraction of image information carry false predictions. And both of them lack explanatory information about the conceptual rules. Our method is able to predict interpretable information in three dimensions: saliency map with concept and rules, which provides more accurate multi-dimensional information, as reflected in the more accurate capture of disease features in image information, clearer concept output, and logical rule information generated based on the predicted concepts.
>
> > W2: "The author considers interpretability on the basis of concept embedding, neural-sybolic reasoning and saliency map. The author lacks the reason why it is necessary to analyze the interpretation from these aspects and whether there are other more dimensions."
>
> **Response**:
>
> **concept**: This approach focuses on understanding the internal representation of high-level concepts in a model.  By analyzing the learned embeddings, researchers can gain insights into how the model associates different input features and concepts.  It helps in understanding the semantic meaning of the model's hidden layers and can reveal if the model has learned meaningful abstractions.
>
> **neural-symbolic**: This combines the power of neural networks with neural symbolic logic, aiming to make models more transparent by grounding their decisions in symbolic rules via concept.  It allows for more explicit concept reasoning processes, which can be more interpretable than purely connectionist models.
>
> **saliency map**: These highlight the parts of the input that are most influential in the model's decision.  By visualizing these maps, one can see which features or pixels are driving the prediction, providing a more intuitive understanding of the model's behavior.
>
> There are also indeed other dimensions:
>
> (1) LIME (Local Interpretable Model-agnostic Explanations)
>
> (2) SHAP (SHapley Additive exPlanations)
>
> These are post-hoc explainability methods, and these techniques could be less scalable, less suitable for large models, and may struggle to align with specific tasks or samples.
>
> Our framework, however, goes beyond these limitations.  By integrating concept-based and attention map explanations, we offer a more intrinsic understanding of the model's decision-making process.  Furthermore, the introduction of neural symbolic methods allows us to delve into clearer concept reasoning and decision-making, thus providing a third dimension of interpretability.
>
> Specifically, we have successfully aligned the attention-based image features with concept-based semantics, ensuring a direct correspondence between the model's perception and the underlying meaning.  Moreover, the neural-symbolic approach enables us to elucidate the interactions among concepts, which is crucial in understanding complex medical scenarios.

---

> > ### Comment · Reviewer_gQUT · 2024-08-10
> > **Thanks for your responses**
> >
> > I appreciate the author's effort to answer my questions. My concerns are well addressed and I will update my score. Btw, I personally have two minor additional questions:
> >
> > 1. What is the concept prediction accuracy in this model？
> >
> > 2. What is the model's robustness against perturbation? This point is also crucial for practical applications in the medical field.

---

> > > ### Author Response · Authors · 2024-08-10
> > >
> > > We sincerely appreciate your prompt and positive response to our rebuttal, along with the improved rating. We are currently conducting experiments to address your additional questions. Thank you for your time!

---

> > > ### Author Response · Authors · 2024-08-12
> > >
> > > 1.  What is the concept prediction accuracy in this model？
> > >
> > > The questions you raised have been very insightful. We used our previous work to perform concept labeling on the DDI dataset. We compared our predicted concepts with the ground truth to determine the accuracy of different backbones, as shown in the table below.
> > >
> > > | Backbone | RN50  | VGG   | DenseNet |
> > > |----------|-------|-------|----------|
> > > | ACC      | 87.12 | 92.13 | 92.83    |
> > >
> > > As can be seen, the concept accuracy predicted by Med-MICN is relatively high, indicating better reliability for the concepts it explains.
> > >
> > > 2. What is the model's robustness against perturbation? This point is also crucial for practical applications in the medical field.
> > >
> > > We conducted stability experiments by applying Gaussian noise with δ=0.1 to the images. We compared the saliency maps generated by our method to those produced by the baseline backbone. Our results show that our method maintains greater stability in focusing on the regions of interest compared to the baseline. Due to the rebuttal policy, we cannot present these results, bu We will present images in the later version.

---

> > > > ### Comment · Reviewer_gQUT · 2024-08-12
> > > > **Discussion**
> > > >
> > > > Thanks for your detailed responses and valuable new experiments. I think it's a solid work and raised my score to 8.

---

### Official Review · Reviewer_WJ93 · 2024-07-08

**Soundness:** 2
**Presentation:** 2
**Contribution:** 2
**Rating:** 5
**Confidence:** 4

**Summary:**

This paper focuses on the explainable classification of medical images with multi-dimensional explanation. The proposed Med-MICN framework contains four modules including feature extraction, auto-annotation, concept embedding, and neural symbolic layer. The incorporation of fuzzy logic rules is novel in the interpretation study of deep neural networks. The experimental results show that Med-MICN surpasses the baseline networks such as vgg and resent, and also outperforms concept bottleneck models (CBM). The generated attention maps and predicted concepts match better than traditional CBM.

**Strengths:**

See summary

**Weaknesses:**

1. The whole presentation of the paper should be further improved including mathematical symbols, figures 2 and 3, and equations. The symbols should be better matched with Figures 2 and 3 to improve readability. It is difficult to understand these figures without reading the contents. Some symbols like Cosine should be given in the figures. Detailed notations are needed in the figures or in the captions.
2. It is better to give pseudo codes of training and inferences to help understand the whole workflow.
3. At the bottom of Page 3, "In instances ... judgment", why do the authors claim alternative explanations can aid judgment? Any references? More explanations may confuse the physicians.
4. In line 131, does $y_{i}$ the one-hot labels?
5. In the Concept labeling alignment module, it is not clear why the authors use "average pooling to heatmaps". I think this will lose some spatial localizations of lesions, for example, one heatmap with multiple lesions and another with one lesion, both of them may have similar average pooling results.
6. Under the line 183, the $V$ is three-dimensional, and why here it has two subscripts.
7. In line 194, does the representation be $c \in \{C_{1}, ..., C_{M}\}$ ?
8. In line 142, the logic symbols should be defined may be in supplementary.
9. Lack of training details. In line 277, how the learning rate decay? Any data augmentations were used?

**Questions:**

See weakness.

**Limitations:**

The whole framework of Med-MICN is complex, and the so-called multi-dimensional explanations are attention maps and predicted concepts which have been proposed in traditional CBM models.

---

> ### Author Rebuttal · Authors · 2024-08-07
>
> We appreciate the reviewers' recognition of our work and your constructive comments. Please find below our detailed responses to the queries you have raised. We hope you could consider increasing our score if our response addresses your concerns :)
>
> > W1: "captions and notations..."
>
> **Response**: We have added notations to Figure 2, 3 and also enhanced the captions for Figures 2, 3, and 4 to improve readability. We have placed the modified table with the notation changes in the PDF. The corresponding original equations have the following modifications:
> - On line 131: N -> M
> - On line 133,135,136: k -> $d_c$
> - On line 130: c = {$p_1, \cdots, p_k$\} -> $\mathcal{C} = ${$C_1, C_2, \dots, C_N $}
> - On line 173: $\mathcal{C} = ${$c_1, c_2, \dots, c_N $} -> $\mathcal{C} = ${$C_1, C_2, \dots, C_N $}
> - Throughout the text, replace all heatmap H with P.
> - On line 184: $H_{p,k,i} = \frac{t_i^T V_{p,k}}{||t_i||\cdot ||V_{p,k}||},\quad p = 1, \ldots , P, \quad k = 1, \ldots , K $ -> $P_{h,w,i} = \frac{t_i^T V_{h,w}}{||t_i||\cdot ||V_{h,w}||},\quad h = 1, \ldots , H, \quad w = 1, \ldots , W $
> - On line 189: $s_i = \frac{1}{H \cdot W} \sum_{h=1}^{H} \sum_{w=1}^{W} P_{h,w,i}$
> - On line 195: $c =$ {$C_1, \ldots, C_M$} -> $c = ${$c_1, \ldots, c_M$}
> - On line 212: $D$ -> $X$ (Here, the dataset and the dimensional representation in the previous text conflict)
> - On line 215: $f_c(x_{m}), \hat{c_m}, f(x_m) = \Theta_c(\Theta_b(x_m))$ for $i \in [M]$ -> $f(x_m) = \Theta_b(x_m), f_c(x_{m}),\hat{c_m} = \Theta_c(f(x_m))$ for $i \in [M]$
> - More detail of $I_{o,j}$ and $I_{r,j}$
>
> > W2: "It is better to give pseudo codes of training and inferences..."
>
> **Response**: The pseudocode has been supplemented, as seen in Figure 2 of the PDF.
>
> > W3: "At the bottom of Page 3..."
>
> **Response**: It has been demonstrated that unidimensional interpretive aids such as saliency map [1,2], concept [3,4], and other information will contribute to medical decision making. To the best of our knowledge, our work is the first to propose a deeper aiding strategy for medical aids using multidimensional interpretable means. The advantage of this multidimensional aid is not only in the cumulative nature of the multiple facilitation strategies but also in the fact that when there is an error in one dimension of interpretability, the information from the other dimensions is able to correct that erroneous information, thus providing more accurately interpreted information.
>
> In addition, we also combine neural symbolic and fuzzy logic to further the connection between the concepts, which makes the interpretation of concepts more auxiliary meaning.
>
> > W4: "In line 131, does $y_i$ the one-hot labels?"
>
> **Response**: Yes, $y_i$ is the one-hot labes.
>
> > W5: "It is not clear why the authors use "average pooling to heatmaps"..."
>
> **Response**:
> |Method|COVID-CT|DDI|
> |--|:---:|:--:|
> |Convolution|**89.81**|80.95|
> |Linear|69.84|73.92|
> |Average pooling|89.79|**91.89**|
>
> Thank you for the valuable suggestions. Our approach is based on experimental results, which show that average pooling is comparable to or even better than other methods.
>
> > W6: "Under the line 183, the $V$ is three-dimensional, and why here it has two subscripts."
>
> **Response**: Yes, it is a 3-dimensional tensor. Here, we use $V_{h,w}$ to represent $V_{h,w,\cdot}$, which is a vector whose length is $D$. The term $ V_{h,w} $ is dot-multiplied with $ t_i^T $ to obtain an intermediate result of size $ \mathbb{R}^{D \times D} $. We then apply pooling to obtain the result for the heatmap at the position $(h, w)$.
>
> > W7: "In line 194, does the representation be ...."
>
> **Response**: The meaning of $c$ is the set of all image concept labels, where $c=${$c_1, . . . , c_M$}. This notation is correct, with each $c_i \in ${$0, 1$} $^N$.
>
> > W8: "In line 142, the logic symbols should be defined may be in supplementary."
>
> **Response**: Thank you. We will define them in supplementary. For example:
> - Negation ($\neg $): This symbol represents the inverse of a truth value. In fuzzy logic, the negation of a degree of truth $x$ is computed as $\neg x = 1 - x$. For example, if $x$ represents a 0.7 probability of a statement being true, then $\neg x$ would be 0.3, indicating a 0.3 probability of the statement being false.
> - T-norm (Conjunction, $\land$): The t-norm, or triangular norm, is a continuous version of the AND operation. It takes two truth values $x$ and $y$ and combines them to produce a new truth value in the range $[0, 1]$. Common t-norms include the minimum function ($\min(x, y)$) and the product function ($x \cdot y$). In the example given, $c_{GO} \land \neg c_{LC}$ would represent the conjunction of "GO" being present and "LC" not being present.
> - T-conorm (Disjunction, $\lor$): The t-conorm, or triangular conorm, is a continuous version of the OR operation. It also takes two truth values $x$ and $y$ and combines them, typically using functions like the maximum function ($\max(x, y)$) or the sum function ($x + y - xy$), ensuring the result is within $[0, 1]$.
>
> > W9: "Lack of training details..."
>
> **Response**: To highlight the significant performance of our method, we did not use learning rate decay; the learning rate was kept constant at 5e-5 throughout the entire training process. For data augmentation, we only used the most basic operations: resize (256), center crop (224), and normalization.
>
> *Reference*
>
> [1] Van der Velden et al. "Explainable artificial intelligence (XAI) in deep learning-based medical image analysis." Medical Image Analysis 79 (2022): 102470.
>
> [2] Patrício et al. "Explainable deep learning methods in medical image classification: A survey." ACM Computing Surveys 56.4 (2023): 1-41.
>
> [3] Patrício et al. "Coherent concept-based explanations in medical image and its application to skin lesion diagnosis." CVPR 2023.
>
> [4] Eminaga et al. "PlexusNet: A neural network architectural concept for medical image classification." Computers in biology and medicine 154 (2023): 106594.

---

> > ### Author Response · Authors · 2024-08-11
> >
> > Dear Reviewer #WJ93,
> >
> > Many thanks for taking the time to review our paper. We have provided our responses to your comments. Can we know whether our responses have addressed your concerns? We are more than happy to hear from you :)
> >
> >
> > Best wishes,
> > Authors of the paper #2244

---

> > ### Comment · Reviewer_WJ93 · 2024-08-14
> > **Response to Rebuttal**
> >
> > Thanks for your detailed rebuttal. All of my concerns have been addressed, and I will increase my score.

---

> > > ### Author Response · Authors · 2024-08-14
> > >
> > > We sincerely appreciate your prompt and positive feedback on our rebuttal, as well as the increase in your rating. Your valuable suggestions have greatly improved the clarity and overall readability of our work.

---

### Official Review · Reviewer_7tbr · 2024-07-09

**Soundness:** 2
**Presentation:** 2
**Contribution:** 4
**Rating:** 6
**Confidence:** 4

**Summary:**

This work introduces Med-MICN, an explainable framework for medical image classification. This framework leverages a concept bottleneck framework combined with a neural symbolic reasoning framework to generate simple explanations for its predictions. In general, strong performance is demonstrated.

**Strengths:**

The overall framework is interesting, and the idea of “multi-dimensional explainability” seems widely applicable. The ability to see some explanation for each concept’s logic seems useful, and the decision rules generated are quite simple. The performance of Med-MICN is quite compelling, with superior accuracy across multiple medical datasets.

**Weaknesses:**

In general, the notation used in the paper is inconsistent and somewhat ill defined. For example, in section 3, N is used to denote the number of sample in a dataset, and k the number of concepts. In section 4, N becomes the number of concepts, and M the number of samples. Lower case k is then used to index along the height dimension of a feature map. Several objects ($I_{o, j}$ and $I_{r,j}$, for example) are not clearly described upon their introduction.

There are a few seemingly key hyper parameters ($\lambda_1$, $\lambda_2$, the threshold used for binarizing concept vectors), but the experimental details do not indicate that there was a validation/development set used to select these values.

If these two points can be addressed, I believe this paper will warrant acceptance.

**Questions:**

There appear to be some notational issues in section 4.2 — initially H and W are used as the height and width of the feature map, but in the definition of $H_{p,k,i}$, it seems that P and K are used to fill the same roll.

“to ensure the truthfulness of 196 concepts, we discard all concepts for which the similarity across all images is below 0.45. “ — does this refer to all training images?

The equation following line 214 is quite confusing — what is the dimensionality of each of the three values returned by $\theta_c$? How does it return $f(x_m)$?

During training, which components are trained jointly? It seems like $L_{neural}$ only applies to the neural symbolic component, while $L_c$ and $L_{task}$ apply only to the concept extraction component and some additional classification fully connected layer.

Figure 4 is somewhat unclear — which cells in the figure should be compared to which?

**Limitations:**

Discussion of limitations is appropriate.

---

> ### Author Rebuttal · Authors · 2024-08-07
>
> We appreciate the reviewers' recognition of our work and your constructive comments. Please find below our detailed responses to the queries you have raised. We hope you could consider increasing our overall score if our response addresses your concerns :)
>
> > W1,Q1: "Notations.."
>
> **Response**: We have placed the modified table with the notation changes in the PDF. The corresponding original equations have the following modifications:
> - On line 131: N -> M
> - On line 133,135,136: k -> $d_c$
> - On line 130: c = {$p_1, \cdots, p_k$\} -> $\mathcal{C} = ${$C_1, C_2, \dots, C_N $}
> - On line 173: $\mathcal{C} = ${$c_1, c_2, \dots, c_N $} -> $\mathcal{C} = ${$C_1, C_2, \dots, C_N $}
> - Throughout the text, replace all heatmap H with P.
> - On line 184: $H_{p,k,i} = \frac{t_i^T V_{p,k}}{||t_i||\cdot ||V_{p,k}||},\quad p = 1, \ldots , P, \quad k = 1, \ldots , K $ -> $P_{h,w,i} = \frac{t_i^T V_{h,w}}{||t_i||\cdot ||V_{h,w}||},\quad h = 1, \ldots , H, \quad w = 1, \ldots , W $
> - On line 189: $s_i = \frac{1}{H \cdot W} \sum_{h=1}^{H} \sum_{w=1}^{W} P_{h,w,i}$
> - On line 195: $c =$ {$C_1, \ldots, C_M$} -> $c = ${$c_1, \ldots, c_M$}
> - On line 212: $D$ -> $X$ (Here, the dataset and the dimensional representation in the previous text conflict)
> - On line 215: $f_c(x_{m}), \hat{c_m}, f(x_m) = \Theta_c(\Theta_b(x_m))$ for $i \in [M]$ -> $f(x_m) = \Theta_b(x_m), f_c(x_{m}),\hat{c_m} = \Theta_c(f(x_m))$ for $i \in [M]$
> - More detail of $I_{o,j}$ and $I_{r,j}$
>
> > W2: "hyperparameters.."
>
> **Response**: Thank you for your valuable feedback. We conducted experiments on reasonable parameter ranges for λ1 and λ2, as detailed in the table below. The table shows the accuracy of various backbones with different combinations of λ1 and λ2. Notably, the combination of λ1=0.1 and λ2=0.1 (used in the paper) performs well overall. When λ1 and λ2 are too small, accuracy generally decreases, indicating that λ1 and λ2 play a positive role in model optimization. Conversely, as λ1 and λ2 increase, different backbones exhibit varying performance. Overall, λ1 and λ2 values between 0.1 and 1 yield good model performance and show significant improvement compared to the baseline.
> | Dataset  | Model | Baseline |  λ1=0.1, λ2=0.1 |  λ1=0.1, λ2=0.01 |  λ1=0.1, λ2=0.05 |  λ1=0.1, λ2=0.5 |  λ1=0.1, λ2=1.0 |  λ1=0.01, λ2=0.1 |  λ1=0.05, λ2=0.1 |  λ1=0.5, λ2=0.1 |  λ1=1.0, λ2=0.1 |
> |-----|---|------|:-------:|:-----:|:-----:|:-----:|:----:|:----:|:---:|:----:|:----:|
> |       | RN50  | 81.36    |    **84.75**    | 83.90 | 82.20 | 82.25 | 83.05 | 81.33 | 83.90 | 83.92 | 82.53 |
> | COVID-CT | DenseNet | 85.59 | 86.44 | 84.75 | 84.74 | **87.29** |      85.59      |       84.09      |       84.75      | 84.12 | 84.28 |
> |       | VGG | 79.60 | 83.05 | 79.66 |       80.52      |      80.51      |      85.60      |       80.85      |       79.66      |      86.44      |    **87.01**    |
>
> We conducted experiments to determine the appropriate threshold values. We sampled 100 samples and the results are as follows:
>
> | Dataset  | 0.1   | 0.2   | 0.3   | 0.4   | 0.45      | 0.5   | 0.6   |
> |----------|-------|-------|-------|-------|-----------|-------|-------|
> | COVID-CT | 59.13 | 63.25 | 76.25 | 84.00 | **90.13** | 88.50 | 86.50 |
>
> >  Q2: "all training images.."
>
> **Response**: We annotated 20% of the training data for fine-tuning the model and labeled all images, including both training and testing images. This is because we need to use the automatically generated concept labels during inference.
>
> > Q3: "equation.."
>
> **Response**: Thank you for pointing this out. The expression in our paper was somewhat unclear. We have revised it as follows:
> For each image $x_m$, after inputting it through the backbone, we obtain $f(x_m)$, represented as $f(x_m) = \theta_b(x_m)$, where $f(x_m) \in \mathbb{R}^{d}$. Next, we input the image feature $f(x_m)$ into the concept encoder $\theta_c$. It is important to note that $\theta_c$ returns two values: one is the predicted concept $\hat{c}_m$ and the other is the concept feature $f_c(x_m)$. The dimensions of these are $\hat{c}_m \in ${$0, 1$}$^N$ $\in \mathbb{R}^{d_c}$ and $f_c(x_m) \in \mathbb{R}^{d_c}$, respectively. In the paper, $d$ and $d_c$ is set to 1000, and $ N $ depends on the number of concepts.
>
> > Q4: "during training.."
>
> **Response**: During training, all modules are trained together. As shown in Table 1, the result of using only the neural-symbolic layer (DCR) is very poor, indicating that joint training is necessary. It is also worth noting that while $\mathcal{L_{neural}}$ primarily affects the output of the neural-symbolic layer, $ \mathcal{L_c} $ impacts the concept prediction results, and $ \mathcal{L}_{task} $ affects the classification results. Due to the nature of backpropagation, these losses also optimize the concept encoder and backbone. As evident from the ablation studies in Table 2, removing any of these losses leads to a decline in overall model performance.
>
> > Q5: "Figure 4.."
>
> **Response**: A full comparison of our method with current saliency and CBM methods in terms of interpretable dimensions and effectiveness is shown in Fig. 4. CBM class methods lack interpretation of saliency map and rules dimensions. saliency map class methods lack interpretation of concept and rules dimensions. For this medical classification task, single dimensions such as saliency map methods do not accurately capture the features of disease images, and CBM-type methods for concept extraction of image information carry false predictions. And both of them lack explanatory information about the conceptual rules. Our method is able to predict interpretable information in three dimensions, saliency map with concept and rules, which provides more accurate multi-dimensional information, as reflected in the more accurate capture of disease features in image information, clearer concept output, and logical rule information generated based on the predicted concepts.

---

> > ### Comment · Reviewer_7tbr · 2024-08-09
> >
> > Thank you for your thorough response. You have addressed the majority of my concerns, and particularly those around notation. My primary concern with the selection of $\lambda_1$ and $\lambda_2$ was that, since they seem to have been selected using test set performance, the models may have been overfit to each test set. However, since they were held constant across datasets after converging on one reasonable set of values, I am somewhat less concerned about this.
> >
> > I think I understand Figure 4 a bit better now, but think the layout could still be improved for clarity. As is, it's a bit confusing how the "saliency map" heading bleeds into the section of the figure from CBM. I think clearer visual separation between areas of the figure corresponding to different methods would improve readability.
> >
> > That said, this is a fairly minor point, and my primary concerns have been addressed. I've increased my score accordingly, and appreciate the authors' response.

---

> > > ### Author Response · Authors · 2024-08-10
> > >
> > > We are profoundly grateful for your timely and positive responses to our rebuttal and the uplift in your rating. Your valuable suggestions have improved the readability of our work. We will make a clearer visual separation between areas of the figure corresponding to different methods.

---

### Official Review · Reviewer_Lw8m · 2024-07-12

**Soundness:** 3
**Presentation:** 2
**Contribution:** 3
**Rating:** 7
**Confidence:** 4

**Summary:**

The authors proposed a novel interpretable model called Med-MICN in this work. This method manages medical image classification tasks with multi-dimensional aspects with neural symbolic and concept semantics. With the help of LLMs, the work performed superior on four medical benchmark datasets. In the ablation study, the authors presented that the performance of the proposed method performed better with the complementary multi-dimensional loss functions in evaluating the classification result.

**Strengths:**

- The paper proposed a novel method that combines extra information (text and logic rules) to enhance the classification performance and the interpretability of the model outputs. This enables the interpretability of the deep learning classification model with multi-dimensional information.
- Instead of post-hoc interpretable methods, the authors integrate fuzzy logic rules into the proposed method. This gives a clear decision rule for image classification and interpretability.
- The method description is relatively clear and easy to follow. The ablation study gave a good overview of the usage of multi-dimensional information.

**Weaknesses:**

- The authors show that the proposed method performed superior to the baseline methods in image classification (Tab. 1). However, there is a missing part on the evaluation of the interpretability. How correct are those heatmaps generated from the proposed model, do they align with the labels or the clinicians?

- Also the comparison of the interpretability between the proposed method and the baseline methods is missing. In Tab. 1, the authors mentioned those methods are interpretable, but they did not evaluate the interpretability. This could be crucial for medical applications.

- For section 4.3, the description of Neural-Symbolic Layer is not easy to follow. What is the difference between the functions of "Concept Polarity" and "Concept Relevance"? It seems like their outputs are the same.

**Questions:**

- The authors used LLM (GPT-4V) in their proposed method. Would it be better or make a difference, if the LLM is a more medical-specific model?

- How do authors deal with different numbers of concept sets from the LLM? If I understood correctly, the LLM could output several different concepts and does not necessarily fit into the input size of the text encoder. (Figure 2.a)

- Did authors try with different $\lambda_1$ and $\lambda_2$?

**Limitations:**

- The proposed method provides an opportunity to interpret the medical image classification results and in the meanwhile also increase the classification performance. However, heatmaps look somehow blurry and may not be trustable (Fig 1).

---

> ### Author Rebuttal · Authors · 2024-08-07
>
> We appreciate the reviewers' recognition of our work and your constructive comments. Please find below our detailed responses to the queries you have raised. We hope you could consider increasing our overall score if our response addresses your concerns :)
>
> > W1, W2: "evaluation of the interpretability"
>
> **Response**: The questions you raised are very insightful. Although quantifying interpretability remains a challenging task with no well-established metrics, our extensive examples demonstrate that the interpretability information provided by our model is meaningful. This is supported by numerous examples presented in our paper, particularly in Figures 1 and 5-12. While aligning saliency maps with features has been a longstanding issue in interpretability, we have compared Med-MICN with baseline saliency maps. As illustrated in Figure 1 of the attached PDF, we used the most distinctive skin modality for clarity. The results show that our method better highlights pathological regions and achieves a significant improvement in alignment compared to the baseline.
>
> > W3: "For section 4.3, the description of Neural-Symbolic Layer is not easy to follow....".
>
> **Response**: *Concept Polarity* focuses on determining the role of a concept as either positive or negative for a given prediction class. The output of the neural network $\Phi_j(\cdot)$ is a soft indicator value between 0 and 1, which signifies the degree to which the concept contributes positively (closer to 1) or negatively (closer to 0) to the prediction. For example, if the output for the concept is 0.8 for class $j$, it suggests that this concept has a strong positive influence on predicting class j.
>
> *Concept Relevance* assesses how useful or significant a concept is within the context of the sample feature for a prediction class. The neural network $\Psi_j(\cdot) $also outputs a soft indicator value between 0 and 1, but this value represents the concept's relevance score, indicating how much it impacts the prediction. If the output for the concept is 0.2 for class $j$, it means that this concept has a relatively low relevance to predicting class j.
>
> While both functions output scalar values between 0 and 1, Concept Polarity indicates the concept's positive or negative influence, and Concept Relevance measures the concept's overall importance or significance in the prediction. These two outputs are combined in Equation (2) to generate the logical reasoning rules for the concept.
>
> > Q1: "The authors used LLM (GPT-4V) in their proposed method. Would it be better..."
>
> **Response**: Thanks to your suggestion, we have fully considered the generalization and accuracy of different diseases when using multimodal models for questioning. Here, we have added two medical multimodal models for comparison experiments (Due to word count limitations, the results from each model will be uniformly presented in the appendix of the paper). The first model is XrayGPT, The XrayGPT model is limited by the form of training data for different kinds of disease images, such as plain view lung CT map (dataset COVID) can not output accurate medical conceptual information. Moreover, for chest CT images, its prediction about different classes of viruses such as COVID output information is also affected and lacks generalization to different disease types.
>
> Besides, we also use the better generalized Llava Med model for comparison. For the COVID dataset, it is capable of obtaining certain medical category information, however, the output disease load categories are not comprehensive compared to the methods used in this framework, and the comprehensiveness of the information.
>
> > Q2: "How do authors deal with different numbers of concept sets from the LLM? ...."
>
> **Response**: It is important to note that each concept within the concept set is processed individually by the text encoder to obtain a corresponding text embedding, rather than inputting the entire concept set simultaneously. The LLM handles varying input sizes through different lengths of concepts, such as "Peripheral ground-glass opacities" and "Bilateral involvement." By obtaining the text embedding for each individual concept, we can derive a concept heatmap for each image and subsequently assign the appropriate concept labels.
>
> > Q3: "Did authors try with different λ1 and λ2?"
>
> **Response**: Thank you for your valuable feedback. We conducted experiments within reasonable parameter ranges for λ1 and λ2, as detailed in the table below. The table shows the accuracy of various backbones with different combinations of λ1 and λ2. When λ1 is fixed, a larger λ2 is preferred, and similarly, when λ2 is fixed, a larger λ1 is preferred. This indicates that each loss function contributes to the model’s predictive accuracy. Additionally, we need to adjust these coefficients for different backbones to achieve optimal performance. For example, with VGG, setting λ1 = 1 and λ2 = 0.1 resulted in a 7.41% increase in accuracy.
>
>
> | Dataset  | Model    | Baseline |  λ1=0.1, λ2=0.1 |  λ1=0.1, λ2=0.01 |  λ1=0.1, λ2=0.05 |  λ1=0.1, λ2=0.5 |  λ1=0.1, λ2=1.0 |  λ1=0.01, λ2=0.1 |  λ1=0.05, λ2=0.1 |  λ1=0.5, λ2=0.1 |  λ1=1.0, λ2=0.1 |
> |----------|----------|----------|:---------------:|:----------------:|:----------------:|:---------------:|:---------------:|:----------------:|:----------------:|:---------------:|:---------------:|
> |          | RN50     | 81.36    |    **84.75**    |       83.90      |       82.20      |      82.25      |      83.05      |       81.33      |       83.90      |      83.92      |      82.53      |
> | COVID-CT | DenseNet | 85.59    |      86.44      |       84.75      |       84.74      |    **87.29**    |      85.59      |       84.09      |       84.75      |      84.12      |      84.28      |
> |          | VGG      | 79.60    |      83.05      |       79.66      |       80.52      |      80.51      |      85.60      |       80.85      |       79.66      |      86.44      |    **87.01**    |

---

> > ### Comment · Reviewer_Lw8m · 2024-08-12
> >
> > I appreciate the authors' effort in answering my concerns. My concerns are majorly well addressed. I have increased my score accordingly.

---

> > > ### Author Response · Authors · 2024-08-14
> > >
> > > We sincerely appreciate your positive feedback on our rebuttal, as well as the increase in your rating. Your valuable suggestions have greatly improved the experimental completeness and overall readability of our work.

---

### Author Rebuttal · Authors · 2024-08-07

We thank the reviewers for their insightful feedback. We have revised and supplemented our work in the following two aspects.

1. **Notations and Captions**: We have added notations to Figure 2, 3 and also enhanced the captions for Figures 2, 3, and 4 to improve readability. We have placed the modified table with the notation changes in the PDF. The corresponding original equations have the following modifications:

- On line 131: N -> M
- On line 133,135,136: k -> $d_c$
- On line 130: c = {$p_1, \cdots, p_k$\} -> $\mathcal{C} = ${$C_1, C_2, \dots, C_N $}
- On line 173: $\mathcal{C} = ${$c_1, c_2, \dots, c_N $} -> $\mathcal{C} = ${$C_1, C_2, \dots, C_N $}
- Throughout the text, replace all heatmap H with P.
- On line 184: $H_{p,k,i} = \frac{t_i^T V_{p,k}}{||t_i||\cdot ||V_{p,k}||},\quad p = 1, \ldots , P, \quad k = 1, \ldots , K $ -> $P_{h,w,i} = \frac{t_i^T V_{h,w}}{||t_i||\cdot ||V_{h,w}||},\quad h = 1, \ldots , H, \quad w = 1, \ldots , W $
- On line 189: $s_i = \frac{1}{H \cdot W} \sum_{h=1}^{H} \sum_{w=1}^{W} P_{h,w,i}$
- On line 195: $c =$ {$C_1, \ldots, C_M$} -> $c = ${$c_1, \ldots, c_M$}
- On line 212: $D$ -> $X$ (Here, the dataset and the dimensional representation in the previous text conflict)
- On line 215: $f_c(x_{m}), \hat{c_m}, f(x_m) = \Theta_c(\Theta_b(x_m))$ for $i \in [M]$ -> $f(x_m) = \Theta_b(x_m), f_c(x_{m}),\hat{c_m} = \Theta_c(f(x_m))$ for $i \in [M]$
- More detail of $I_{o,j}$ and $I_{r,j}$

2. **Hyperparameters and Experimental Details**: We conducted experiments on reasonable parameter ranges for λ1 and λ2, as detailed in the table below. The table shows the accuracy of various backbones with different combinations of λ1 and λ2. Notably, the combination of λ1=0.1 and λ2=0.1 (used in the paper) performs well overall. When λ1 and λ2 are too small, accuracy generally decreases, indicating that λ1 and λ2 play a positive role in model optimization. Conversely, as λ1 and λ2 increase, different backbones exhibit varying performance. Overall, λ1 and λ2 values between 0.1 and 1 yield good model performance and show significant improvement compared to the baseline.

| Dataset  | Model    | Baseline |  λ1=0.1, λ2=0.1 |  λ1=0.1, λ2=0.01 |  λ1=0.1, λ2=0.05 |  λ1=0.1, λ2=0.5 |  λ1=0.1, λ2=1.0 |  λ1=0.01, λ2=0.1 |  λ1=0.05, λ2=0.1 |  λ1=0.5, λ2=0.1 |  λ1=1.0, λ2=0.1 |
|----------|----------|----------|:---------------:|:----------------:|:----------------:|:---------------:|:---------------:|:----------------:|:----------------:|:---------------:|:---------------:|
|          | RN50     | 81.36    |    **84.75**    |       83.90      |       82.20      |      82.25      |      83.05      |       81.33      |       83.90      |      83.92      |      82.53      |
| COVID-CT | DenseNet | 85.59    |      86.44      |       84.75      |       84.74      |    **87.29**    |      85.59      |       84.09      |       84.75      |      84.12      |      84.28      |
|          | VGG      | 79.60    |      83.05      |       79.66      |       80.52      |      80.51      |      85.60      |       80.85      |       79.66      |      86.44      |    **87.01**    |

We conducted experiments to determine the appropriate threshold values. We sampled 100 samples and the results are as follows:

| Dataset  | 0.1   | 0.2   | 0.3   | 0.4   | 0.45      | 0.5   | 0.6   |
|----------|-------|-------|-------|-------|-----------|-------|-------|
| COVID-CT | 59.13 | 63.25 | 76.25 | 84.00 | **90.13** | 88.50 | 86.50 |

---

### Decision · Program_Chairs · 2024-09-25

**Decision:**

Accept (poster)

**Comment:**

After discussion, all reviewers agree to accept this paper.
The authors are encouraged to further improve this work by addressing feedback from the reviewers, in particular, the following:

1. Add evaluation of the generated heatmaps
2. Compare the interpretability between the proposed method and the baseline methods
3. Improve the notations